# Emulation of coherent absorption of Fock-state quantum light in a programmable linear photonic circuit

Govind Krishna [1] ✉, Jun Gao [1,2] ✉, Sam O'Brien[1], Rohan Yadgirkar[1], Venkatesh Deenadayalan[3], Stefan Preble[3], Val Zwiller [1] & Ali W. Elshaari [1] ✉

Non-Hermitian quantum systems, governed by nonunitary evolution, offer powerful tools for manipulating quantum states through engineered loss. A prime example is coherent absorption, where quantum states undergo phase-dependent partial or complete absorption in a lossy medium. Here, we demonstrate a fully programmable implementation of nonunitary transformations that emulate coherent absorption of quantum light using a programmable integrated linear photonic circuit, with loss introduced via coupling to an ancilla mode. Probing the circuit with a single-photon dual-rail state reveals phase-controlled coherent tunability between perfect transmission and perfect absorption. A two-photon NOON-state input, by contrast, exhibits switching between deterministic single-photon absorption and probabilistic two-photon absorption. Across a broad range of input phases and circuit configurations, we observe nonclassical effects including anti-coalescence and bunching, together with continuous and coherent tuning of output Fock-state probability amplitudes. Classical Fisher information analysis reveals phase sensitivity peaks of 1 for single-photon states and 3.4 for NOON states, exceeding the shot-noise limit of 2 and approaching the Heisenberg limit of 4 for two-photon states. The experiment integrates quantum state generation, programmable photonic circuitry, and photon-number-resolving detection, establishing ancilla-assisted circuits as powerful platforms for programmable quantum state engineering, filtering, multiplexed sensing, and nonunitary quantum simulation.

Non-unitary transformations describe irreversible processes, such as loss, gain, or measurement back-action in open systems, in contrast to the reversible unitary dynamics of isolated systems[1]. In quantum photonics, unitary transformations preserve photon number and are typically implemented via multiport interferometer networks[2,3]. Extending linear optics to the non-unitary domain enables modeling dissipation, decoherence, amplification, and generalized quantum

measurements, such as positive operator-valued measures (POVMs), as well as imperfections, and facilitates probabilistic quantum information protocols[4–7].

Any linear transformation, including non-unitary ones, can be realized by embedding it into a larger unitary on an extended Hilbert space with ancilla modes representing the environment. Early frameworks by Knöll and co-workers[8,9] and by He and colleagues[10]

[1]Department of Applied Physics, KTH Royal Institute of Technology, Stockholm, Sweden. [2]School of Optical and Electronic Information, Huazhong University of Science and Technology, Wuhan, China. [3]Department of Electrical and Microelectronic Engineering, Rochester Institute of Technology, Rochester, NY 14623, USA. ✉e-mail: govindk@kth.se; jungao@hust.edu.cn; elshaari@kth.se

demonstrated this approach, but could not address transformations involving both loss and gain. Tischler and co-workers[11] later introduced a quasiunitary dilation method that embeds general linear transformations into physically realizable optical networks using ancilla modes and, when necessary, parametric amplification. A paradigmatic instance of a non-unitary optical device is a lossy beam splitter, a 2 × 2 interferometer with an intrinsic internal absorption coefficient. Such beam splitters exhibit highly tunable, apparently nonlinear quantum interference effects[7,12–14]. A lossy beam splitter underpins the non-unitary process of coherent absorption, where a lossy medium partially or fully absorbs coherent light incident from both sides, with the absorption depending not only on the medium, but also on the relative phase and amplitudes of the input light. As illustrated schematically in Fig. 1a, classically coherent absorption arises when counter-propagating waves form a standing wave, with the absorption determined by the absorber's position within the resulting field profile. Complete absorption or coherent perfect absorption (CPA) occurs at antinodes, full transmission at nodes, and partial absorption elsewhere. Even though CPA refers to complete absorption, in this manuscript, we will use the abbreviation CPA to refer generically to coherent absorption processes, including both partial and complete absorption, unless explicitly stated otherwise. CPA, proposed by Chong et al.[15] as the time-reversed analogue of a laser, was first experimentally demonstrated using silicon slabs by Wan et al.[16]. Since then, classical CPA has been studied for various applications, including light-by-light control[17], coherent optical switching[18], signal modulation[19,20], dark pulse generation[21], and all-optical coherent amplification[22].

In the quantum regime, coherent absorption extends naturally to Fock-based quantum states of arbitrary photon number. The input field can be expressed as a superposition of two orthogonal effective modes that play roles analogous to the cosine (bright) and sine (dark) modes of the field-one that couples to the absorbing channel and another that remains lossless[23], as depicted in Fig. 1a. The relative phase between the input modes determines the amplitude coefficients of the quantum state in this new basis, and thereby controls the effective probability of absorption. For single-photon quantum states, these effective modes are symmetric and antisymmetric path superpositions of the input modes, corresponding to the single-photon schematic shown in Fig. 1b, whereas for multiphoton NOON states, the relative input phase defines the decomposition of the total state into photon-number components of the bright and dark modes, as illustrated for the two-photon case in Fig. 1c, thereby controlling the likelihood of single- or multi-photon absorption events.

Single photon CPA has been experimentally demonstrated in subwavelength films[24,25], where a photon prepared in a balanced path superposition can be deterministically absorbed or transmitted depending on the input relative path phase. With multiphoton inputs, such as two-photon NOON states, CPA induces nonlinear-like behavior even in linear media[23,26–28]. Theoretical models show that lossy beam splitters can selectively absorb certain superpositions and suppress or enhance specific Fock components at the output[7,23,28]. This provides mechanisms for state-selective filtering[13,23,29,30], coherent manipulation of photon correlations[7,12,26,27,31], and realization of the anti-Hong-Ou-Mandel (Anti-HOM) effect[32], with both local[24,25] and nonlocal[26,28] control over photon absorption.

Despite significant progress, previous quantum CPA experiments relied on static, unprogrammable components. Absorber parameters were fixed, allowing only limited operating points, while input phase control required moving parts, limiting stability and tuning speed. Motivated by the integrated-photonic analogue shown in Fig. 1, in this work, we overcome these constraints by emulating quantum coherent absorption in a programmable integrated photonic circuit. Our architecture embeds a tunable lossy beam splitter transformation into an 8-mode universal interferometer circuit based on Clements

architecture[3]. The circuit is synthesized using the quasi-unitary decomposition method proposed by Tischer et al. in ref. 11 and allow full tunability of all CPA parameters, such as the input state phase $\phi$, and the reflectivity $r$, transmissivity $t$, absorptivity $A$, and the internal phase $\phi_{rt}$ of the lossy beam splitter. We probe the circuit using single-photon balanced dual-path superposition states and two-photon NOON states, corresponding to the quantum implementations schematically summarized in Fig. 1b, c. The programmable architecture allows access to all previously reported quantum CPA effects, including deterministic absorption, anti-coalescence, photon bunching, and probabilistic two-photon absorption, within a single device. For the two-photon NOON-state experiments, the measured output photon-count statistics exhibit high-fidelity agreement with theory, with Bhattacharyya overlaps (a measure of state similarity, where 1 indicates a perfect match) exceeding 0.93 across all configurations, and average overlap values of 0.9850 ± 0.0122 and 0.9910 ± 0.0092 for the two lossy beam splitter configurations investigated in this manuscript, while simultaneously enabling programmable redistribution of phase sensitivity among the output Fock states. Our results demonstrate programmable ancilla-assisted photonic circuits as practical tools for quantum state engineering, non-unitary quantum simulations, quantum state filtering and adaptive, reconfigurable, and multiplexed quantum sensing.

## Results

### Implementation of non-unitary transformation

In this work, we consider port-symmetric lossy beam splitters with identical coefficients for both ports, and such beam splitters are described by a non-unitary 2 × 2 scattering matrix $S$[7]:

$$\begin{pmatrix} \widehat{a}_{\text{out}} \\ \widehat{b}_{\text{out}} \end{pmatrix} = S \begin{pmatrix} \widehat{a}_{\text{in}} \\ \widehat{b}_{\text{in}} \end{pmatrix}, S = \begin{pmatrix} t & r \\ r & t \end{pmatrix}, \tag{1}$$

where $t = |t|e^{i\phi_t}$ and $r = |r|e^{i\phi_r}$ are the complex transmission and reflection amplitudes. Throughout this work, we define the internal phase as $\phi_{rt} = \phi_r - \phi_t = \phi_r$ and $\phi_t = 0$. The light absorption is quantified by the intrinsic absorption coefficient $|A|^2 = 1 - |t|^2 - |r|^2$. See the "Methods" section "Lossy beam splitter constraints" for additional constraint equations governing these parameters. Although coherent absorption can also occur in reciprocal but port-asymmetric beam splitters, we focus on the port-symmetric case as it isolates the simplest regime, where CPA is realized through balanced two-port interference without loss of generality.

Figure 2a illustrates the workflow used to cast a 2 × 2 non-unitary lossy beam splitter matrix (Eq. 1) into a larger unitary matrix, enabling its implementation using a linear photonic circuit. We adopt a quasiunitary extension strategy[11], embedding the lossy transformation into a larger unitary matrix by introducing ancillary vacuum modes (see the "Methods" section "Quasi-unitary decomposition scheme"). We numerically confirmed that, for all beam splitter configurations studied, a single ancilla mode suffices, leading to a compact three-mode implementation of the CPA circuit.

This augmented transformation is then physically synthesized using a rectangular mesh of Mach–Zehnder interferometers (MZIs) via the Clements decomposition scheme[3]. Each MZI consists of two fixed 50:50 beam splitters, an internal phase shifter $\theta$ and an external phase shifter $\phi$, together enabling implementation of arbitrary SU(2) transformations. The decomposition determines the phase settings $(\theta_i, \phi_i)$ of each MZI required to implement the full 3 × 3 unitary matrix (see the "Methods" section "Clements decomposition scheme").

In this work, we consider circuits emulating two classes of lossy beam splitters characterized by different constraints on their internal parameters. Type 1 maintains a fixed internal phase $\phi_{rt} = \pi$, while the magnitude ratio $|r|/|t|$ varies from 0 to 1 as the absorption coefficient $|A|^2$ increases from 0 to 0.5. Type 2 imposes symmetry ($|t| = |r|$), which

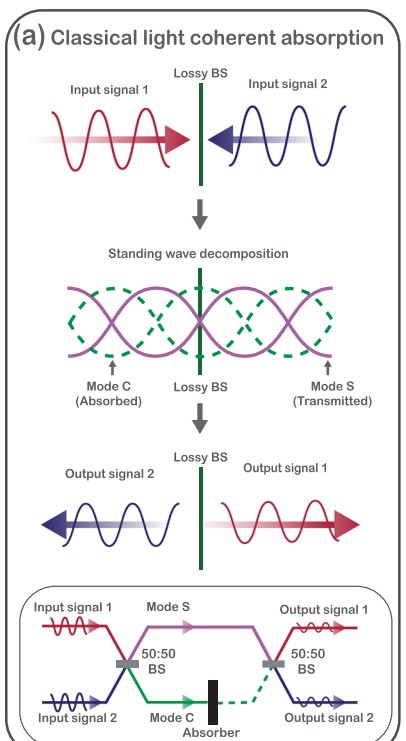

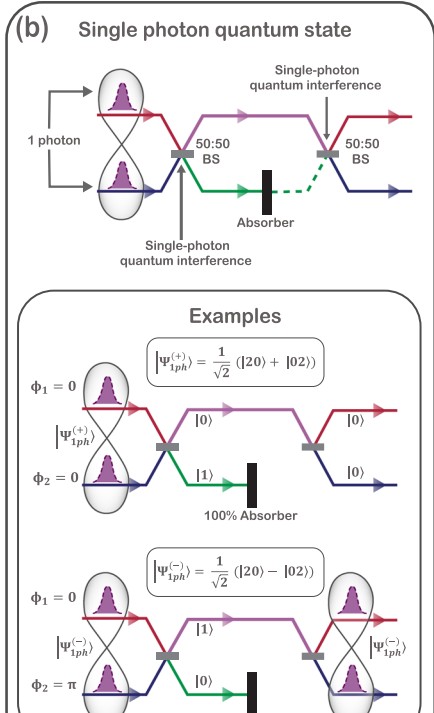

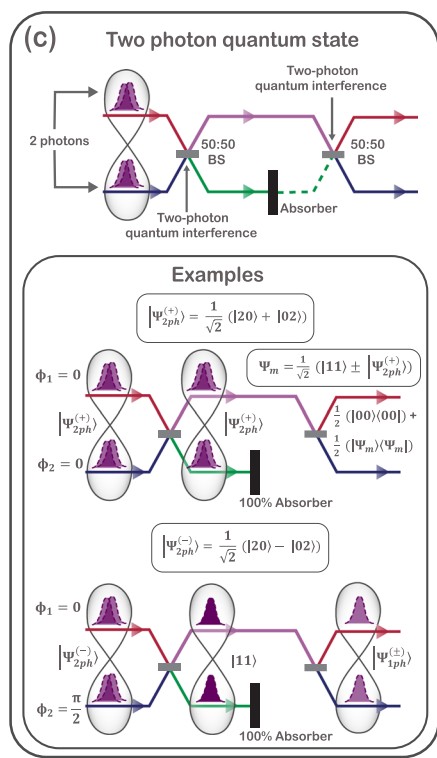

**Fig. 1 | Classical and quantum coherent absorption in an integrated photonic analogue. a** Classical coherent absorption at an effective lossy beam splitter. Two counter-propagating input fields interfere to form a standing wave, which can be decomposed into a superposition of two components: an absorbing mode (mode C) that couples to the loss channel and a non-absorbing mode (mode S) that is fully transmitted. The relative weight of these two components is controlled by the phase difference between the input fields. This standing-wave picture motivates the integrated-photonic analogue (bottom), where a balanced interferometric network maps traveling-wave inputs onto standing-wave supermodes and implements absorption by coupling only mode C to an absorber. **b** Single-photon implementation: a single photon prepared in a coherent superposition across the two input modes undergoes single-photon quantum interference in the network, enabling phase-controlled coupling to the absorber (illustrative examples shown). **c** Two-photon implementation: two-photon quantum interference (e.g., NOON-type inputs) enables phase-dependent coherent absorption of multiphoton states (illustrative examples shown).

causes $\phi_{rt}$ to vary from $\pi/2$ to $\pi$ over the same range of $|A|^2$. These two classes illustrate complementary pathways for tuning coherent absorption through independent control of the beam splitter amplitude and phase parameters.

**Experimental architecture**

Figure 2b depicts our experimental architecture alongside a conceptual comparison to free-space CPA using a static lossy beam splitter[24,26]. The core transformation is implemented by three reconfigurable MZIs (MZI$_1$–MZI$_3$). Unlike static beam splitters in a free-space setup, our circuit allows tunability of $|t|$, $|r|$, $A$, and $\phi_{rt}$.

From a theoretical mapping between the target unitary $S_{\text{total}}$ and the MZI mesh scattering matrix, we derive compact relations for the MZI phase shifts:

$$\theta_{\text{MZI}_2} = 2\cos^{-1}\left(\sqrt{2|A|^2}\right), \tag{2}$$

$$
\begin{aligned}
\phi_{\text{MZI}_3} - \phi_{\text{MZI}_2} &= \arg\left(\frac{t+r}{t-r}\right) + \frac{\theta_{\text{MZI}_2}}{2} + \frac{\pi}{2} \\
&= \text{sgn}(\cos\phi_{rt})\tan^{-1}\left(\frac{2|t||r|\sin(\phi_{rt})}{|t|^2 - |r|^2}\right) + \frac{\theta_{\text{MZI}_2}}{2} + \frac{\pi}{2},
\end{aligned}
\tag{3}
$$

where the sign function is defined as

$$
\text{sgn}(x) = \begin{cases} +1, & x > 0, \\ -1, & x < 0. \end{cases}
$$

These expressions are consistent with the MZI decomposition schemes and were verified numerically, with detailed derivations presented in Supplementary Note 2. We fix $\theta_{\text{MZI}_1} = \theta_{\text{MZI}_3} = 0.5\pi$ (50:50 splitters) and $\phi_{\text{MZI}_1} = \pi$, as these phase values remain the same across the decomposition of all beam splitter configurations. All other on-chip MZIs are set to the passive waveguide modes $\theta = \pi$ (bar state) or $\theta = 0$ (cross state), depending on the desired light routing through the circuit. As evident from Eq. 2, $\theta_{\text{MZI}_2}$ sets the absorption coefficient $|A|^2$ of the emulated beam splitter, controlling the fraction of light diverted to the ancilla output (mode 3).

We probe this circuit with two classes of quantum input states: single-photon balanced path superposition states and two-photon NOON states. Both states are initialized using a single tunable MZI (depicted on the bottom panel of Fig. 2b, configured to perform a balanced 50:50 beam splitter operation.

For the single-photon case, an input photon is split into a superposition across the two signal modes that form the inputs to the circuit:

$$|\Psi_{1ph}\rangle = \frac{1}{\sqrt{2}}\left(e^{i\phi}|10\rangle - |01\rangle\right), \tag{4}$$

For the 2-photon NOON-state case, two indistinguishable photons injected into the same MZI produce another path-entangled state:

$$|\Psi_{\text{NOON}}\rangle = \frac{1}{\sqrt{2}}\left(e^{2i\phi}|20\rangle - |02\rangle\right). \tag{5}$$

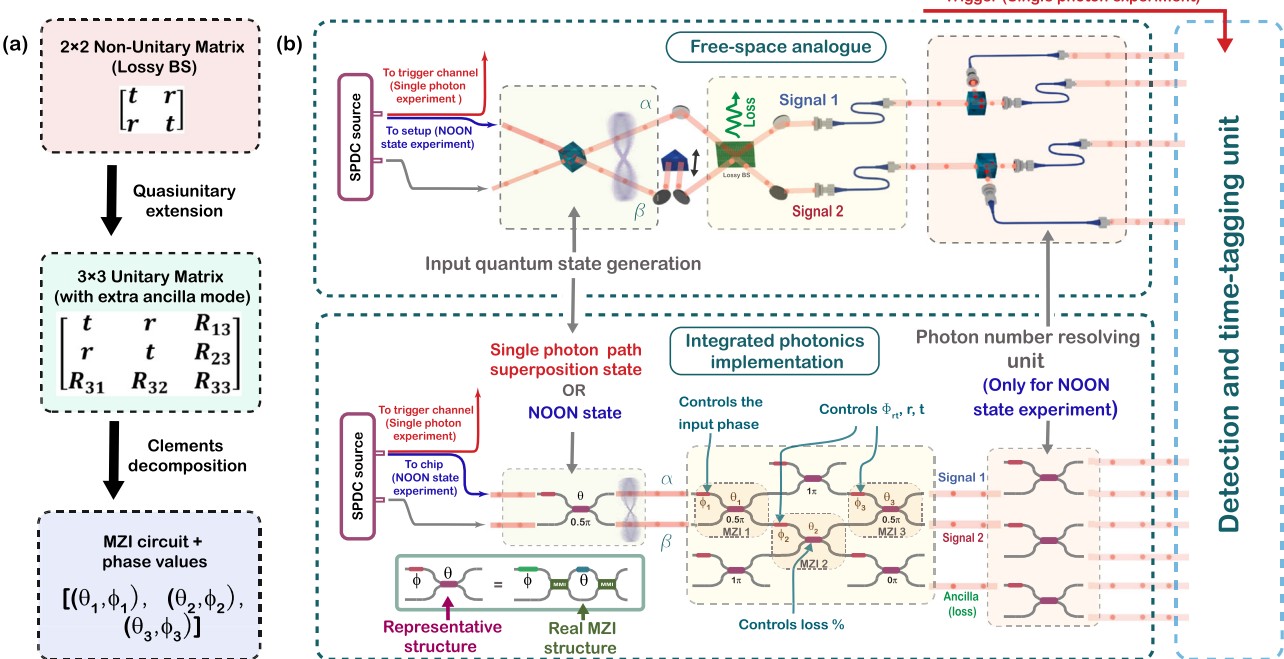

**Fig. 2 | Schematic overview of the coherent perfect absorption experiment.**
**a** Schematic of the theoretical synthesis workflow: a 2 × 2 non-unitary scattering matrix is embedded into a 3 × 3 unitary via quasiunitary extension and decomposed into a Mach–Zehnder interferometer mesh using the Clements scheme. $R_{ij}$ represents the additional complex numbers resulting from the quasiunitary decomposition step. **b** Conceptual comparison between free-space and integrated photonic implementations of coherent absorption. In both cases, the input is a quantum superposition over two spatial modes (a single-photon state or a two-photon NOON state). This state is prepared using a bulk 50:50 beam splitter in the free-space implementation or an on-chip Mach–Zehnder interferometer in the

integrated photonic platform, following generation of indistinguishable photon pairs by an external SPDC source. In the free-space case, this state interacts with a static, lossy beam splitter (Green coloured plane in the top panel), with absorption governed by the relative phase of the input state. The integrated version emulates this process using a programmable 3-mode interferometer (MZI₁–MZI₃), enabling tunable control over $|r|$, $|t|$, $\phi_{rt}$, and A of the emulated beam splitter. All other on-chip MZIs are configured as passive waveguides to route light without affecting the transformation circuit. Detection is performed via heralding (single-photon state case) or photon-number-resolving coincidence counting (NOON state case).

In both cases, the relative mode phases $\phi$, $2\phi$ control the interference with the effective lossy beam splitter (here, and throughout, $\phi/2\phi$ denotes the input-state phase and should not be confused with the internal beam splitter phase $\phi_{rt}$). The quantum state phase $\phi$ is applied on top of a fixed $\pi$ offset of $\phi_{MZI_1}$. This $\pi$ phase arises from the lossy beam splitter matrix decomposition and is common to all lossy beam splitter configurations implemented in this work, whereas $\phi$ (or $2\phi$ for the NOON state) denotes the externally controlled input-state phase applied to that mode. The doubled phase dependence ($2\phi$) in the case of the NOON state enables enhanced phase sensitivity.

Figure 3a illustrates the complete experimental setup used in this work. Correlated photon pairs are generated via type-II spontaneous parametric down-conversion (SPDC) in a periodically poled KTP (PPKTP) crystal, and the down-converted photons, centered around 1570 nm, are collected into single-mode fibers and routed to the input of the integrated photonic chip.

The chip as a whole comprises an 8 × 8 interferometer mesh based on the Clements architecture[3], and for this work, we employ three functional subnetworks: the single-MZI state preparation unit, the central 3-mode CPA core, and a photon-number-resolving MZI array for output analysis, as shown in Fig. 3b. Figure 3c shows the Hong-Ou-Mandel interference curve measured at the state-preparation MZI used to generate two-photon NOON states for the NOON state experiment (details and results explained in a later section).

All output modes are connected to superconducting nanowire single-photon detectors (SNSPDs), and each detector is assigned to a separate channel of a Swabian time-tagger, which records and processes detection events with sub-nanosecond timing resolution. Coincidence measurements across all output channels are performed

simultaneously to reconstruct output statistics. In the NOON state experiment, an additional photon-number-resolving step is implemented by routing each of the three output modes from the chip through a fixed 50:50 MZI, effectively splitting each mode into two detection channels (Figs. 2b and 3a). This enables coincidence detection across all $\binom{6}{2} = 15$ detector combination pairs, from which two-photon output statistics are reconstructed. For output Fock states containing two photons in a single mode (e.g., $|200\rangle$, $|020\rangle$, $|002\rangle$), the measured coincidence counts are multiplied by 2 to account for the 50 % probability that two photons entering the photon-number-resolving MZI are split into a $|11\rangle$ output, rather than remaining in a $|20\rangle$ or $|02\rangle$ state, which yields no coincidence detection. In contrast, for the single-photon experiment, only one photon from the SPDC pair is injected into the chip, while the other serves as a heralding trigger. Coincidence events between the herald and the three chip outputs are recorded, yielding three coincidence channels. This detection strategy provides full access to phase-dependent transmission, absorption, and redistribution effects in both lossy beam splitter configurations.

A 2 ns coincidence window was employed for all coincidence counts measurements presented in this manuscript. See the "Methods" section "The photonic chip, thermal stabilzation and control" and Supplementary Note 3 for additional information on the experimental setup.

**Single-photon state $\left(|\Psi_{1ph}\rangle\right)$ experiment**
Figure 4 presents measurements with the $\left|\Psi_{1ph}\right\rangle$ state defined in Eq. 4, injected into the two types of beam splitter configurations: type 1

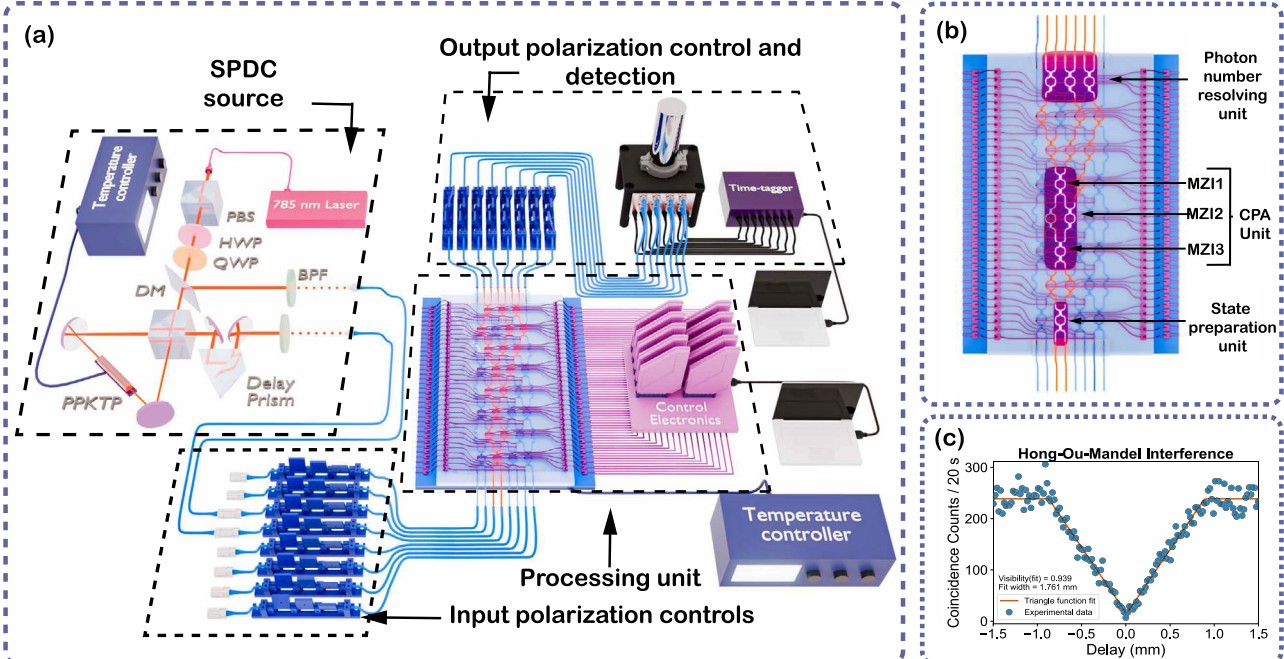

**Fig. 3 | Experimental setup and photonic circuit layout. a** Schematic of the experimental setup. Photon pairs are generated via type-II spontaneous parametric down-conversion (SPDC) in a PPKTP crystal pumped by a 785 nm CW laser. Light is coupled in and out of the chip using edge couplers. Fiber-based polarization controllers are used before the input to ensure efficient coupling to the TE mode of the waveguides, and after the chip to rotate the output light to the polarization for which the detectors show maximum efficiency. Output photons are detected using superconducting nanowire single-photon detectors (SNSPDs) (in a 2.6 K cryostat), and coincidence counts are recorded with a Swabian time-tagger. **b** Functional layout of the 8 × 8 programmable photonic chip used in the experiment. The mesh comprises three active subnetworks: (i) a single-MZI unit for preparing single-photon superposition or NOON states, (ii) a central three-mode programmable interferometer implementing the CPA transformation, and (iii) a photon-number-resolving array of MZIs for state analysis. **c** On-chip Hong-Ou-Mandel interference corresponding to NOON state preparation, fitted using a triangular function $y = a - b|x - x_0|$, shows a visibility of 0.939. This interference dip is obtained by tuning the relative delay between the photon pair paths using a prism mounted on a translational stage on one SPDC output path.

(fixed $\phi_{rt} = \pi$), and type 2 ($|t| = |r|$). The single-photon character of the input was verified via a heralded second-order correlation measurement (see the "Methods" section "The SPDC source" for more details), yielding $g^{(2)}(0) = 0.014$ and a maximum $g^{(2)}(\tau)$ of only 0.027 within the 2 ns coincidence window. Panels (a–c) and (d–f) respectively correspond to these cases, showing the output intensities at the two signal ports (S1, S2) and the ancilla port (Anc) as functions of input quantum state phase $\phi$ and $|A|^2$.

The observed intensity oscillations arise from coherent single-photon quantum interference at $MZI_1$ and $MZI_3$. $MZI_1$ effectively projects the input state onto a basis of two orthogonal superpositions-the antisymmetric state $|1_-\rangle = \frac{1}{\sqrt{2}}(|10\rangle - |01\rangle)$ propagating through the upper arm and the symmetric state $|1_+\rangle = \frac{1}{\sqrt{2}}(|10\rangle + |01\rangle)$ through the lower arm, and their amplitudes vary as $\cos\phi$ and $\sin\phi$, respectively. For $\phi = 0, 2\pi,...$ ($|1_-\rangle$), the photon exits deterministically through the upper arm of $MZI_1$ and reaches the signal ports without absorption. For $\phi = \pi, 3\pi,...$ ($-|1_+\rangle$), the photon exits the lower arm, and gets routed to the ancilla via $MZI_2$, with a probability determined by the programmed loss (the set $|A|^2$ value). Unabsorbed photons are routed to the bottom input of $MZI_3$. Thus, absorption peaks occur at symmetric input phases. Nearly perfect absorption (~100%) is observed at $\phi = \pi$ for $|A|^2 = 0.5$ (Fig. 4b, e). The oscillations follow a $2\pi$ periodicity in the input phase. A more detailed theoretical framework of the quantum state evolution in the circuit is provided in Supplementary Note 4 (Subsection 4.1)

If the input state phase $\phi$ is set independently before $\phi_{MZI_1}$, (noting that in the present implementation $\phi_{MZI_1}$ incorporates both the fixed $\pi$ phase arising from the lossy beam splitter decomposition and the externally applied input-state phase) applying an additional $\pi$ phase shift to $\phi_{MZI_1}$ (so that the total phase on that phase shifter becomes $2\pi$) swaps the absorbed and transmitted states: the antisymmetric state is absorbed, while the symmetric state is transmitted. For any general

input phase $\phi$, the input state projects onto both symmetric and anti-symmetric modes, resulting in phase-dependent routing between signal and ancilla ports. This behavior is evident in panels Fig. 4b, e. The experimental data agree well with the theoretical predictions. Other values of $\phi_{MZI_1}$ (if the state phase $\phi$ is set independently before $\phi_{MZI_1}$) program the circuit to absorb a chosen intermediate superposition state and transmit its orthogonal counterpart. This enables programmable quantum state filtering, selectively routing one superposition component to the ancilla without actually losing the photon.

The internal phase $\phi_{rt}$ (of the emulated lossy beam splitter) governs the relative phase difference between the two inputs to $MZI_3$ (as described by Eq. 3), thereby modulating the interference condition at that MZI. This interference determines how the unabsorbed component of the input state is distributed between the two signal ports. Figure 4g shows that signal port interference visibility increases with absorption for type 1 but remains fixed at ~100% for the symmetric type 2 configuration. The relative phase shift between the S1 and S2 output intensities, plotted in Fig. 4h, stays zero for type 1 due to its fixed $\phi_{rt} = \pi$, while for type 2 it evolves continuously with $|A|^2$. The observations in both Fig. 4g, h are direct consequences of the $\phi_{rt}$-dependent interference at $MZI_3$ (a detailed theoretical framework of this interference is discussed in Supplementary Note 4 (Subsection 4.4)).

To quantify phase sensitivity, we evaluate the classical Fisher information (FI) for each detected output Fock state $|\psi_i\rangle$. For a given lossy BS configuration, the maximum FI with state $|\psi_i\rangle$ across all input phase values is defined as:

$$F_{|\psi_i\rangle}^{max} = \max_\phi \left[ \frac{1}{P_{|\psi_i\rangle}(\phi)} \left( \frac{dP_{|\psi_i\rangle}(\phi)}{d\phi} \right)^2 \right], \tag{6}$$

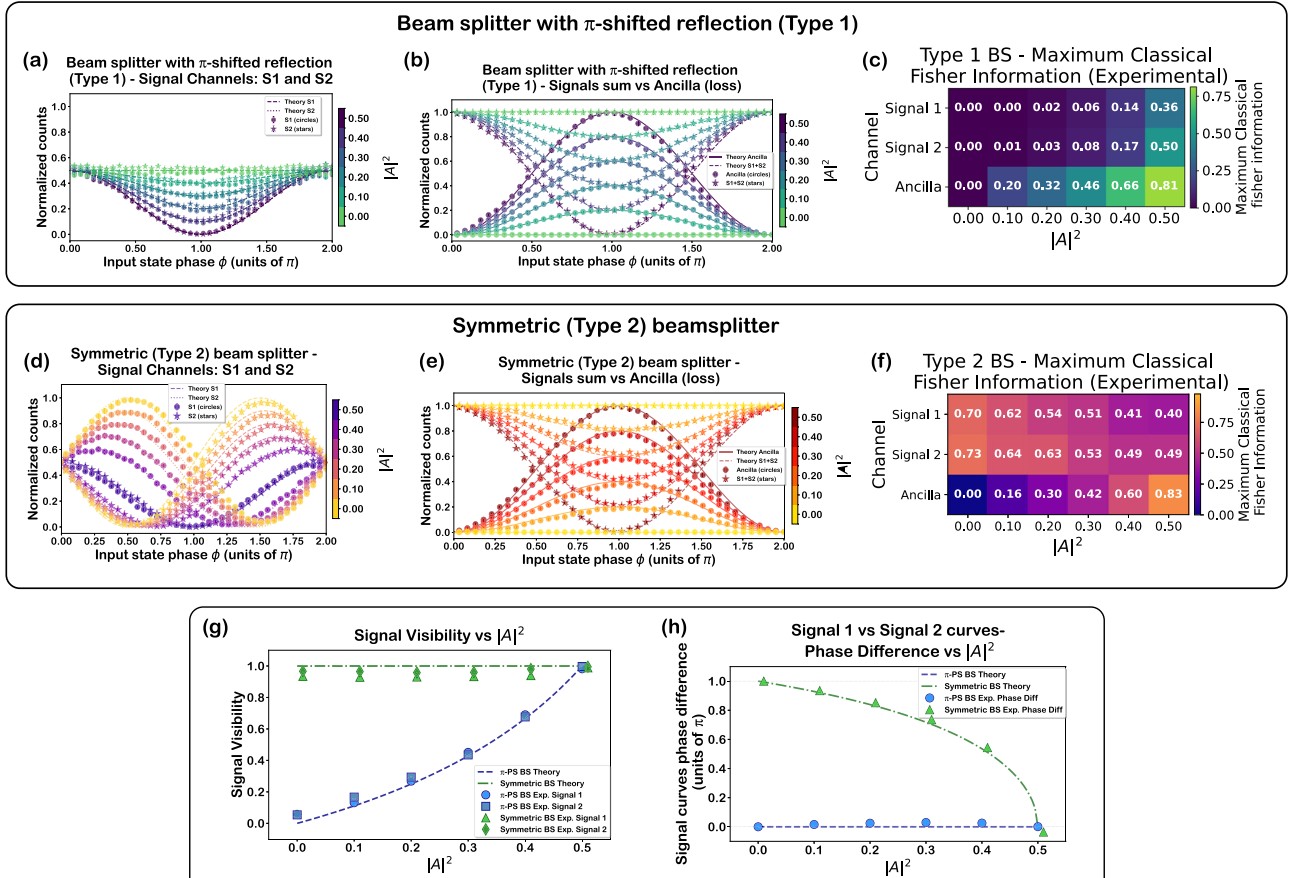

**Fig. 4 | Single-photon state experiment results. a–c** Normalized counts measured at signal (S1, S2) and ancilla (Anc) ports as functions of input quantum state phase $\phi$, with the intrinsic absorption coefficient $|A|^2$ represented by the color scale, for type 1 ($\phi_{rt} = \pi$) beam splitter. For type 1, the MZI$_3$ input phase difference is fixed to $\Delta\phi_{in} = 0$ (see Eq. 3 and Supplementary Note 4 (Subsection 4.4)). Panels show: **a** individual signal ports; **b** signals sum (S1 + S2) vs. Anc; **c** corresponding Fisher information per output mode for different absorption settings (see Eq. 6). Same as above for type 2 symmetric beam splitter ($|t| = |r|$), with **d** signal outputs; **e** Signals sum vs. Anc; **f** Fisher information. For type-2, the MZI$_3$ input phase difference is fixed to $\Delta\phi_{in} = \pi/2$ (see Eq. 3 and Supplementary Note 4 (Subsection 4.4)). Each data point corresponds to coincidence counts between the circuit outputs and a heralding detector triggered by the twin photon from the SPDC source. All data for both beam splitter configurations are normalized by the total coincidences across all outputs and corrected for variations in detector efficiency and coupling efficiency of the chip's output waveguides to the corresponding fibers. Error bars represent one standard deviation, calculated as $\sqrt{N}$ from Poisson counting statistics, with proper propagation through the normalization procedure. **g** Phase-dependent visibility of signal outputs vs. absorption; **h** relative phase shift between S1 and S2 intensities. The lines denote theoretical predictions; symbols are experimental data. The theoretical data are obtained by applying the CPA unitary transformation matrix $S_{total}$ to the input state vector and computing output intensities.

where $P_{|\psi_i\rangle}(\phi)$ is the probability of detecting the output Fock state $|\psi_i\rangle$[33,34]. The classical Fisher information quantifies the sensitivity of the output state probability to the phase of the input quantum state. The total Fisher information at a given input phase $\phi$ and lossy BS configuration is obtained by summing the contributions of all accessible output Fock states:

$$F_{tot}(\phi) = \sum_i \frac{1}{P_{|\psi_i\rangle}(\phi)} \left(\frac{dP_{|\psi_i\rangle}(\phi)}{d\phi}\right)^2. \tag{7}$$

The maximum phase sensitivity of the circuit is then determined from the peak value of $F_{tot}(\phi)$ over $\phi$.

The Fisher information provides a natural measure of phase sensitivity because it quantifies how strongly the output detection probabilities $P_{|\psi_i\rangle}(\phi)$ respond to small changes in the input phase. A larger FI indicates that even a slight variation of $\phi$ produces a measurable change in the photon-count statistics. For a fixed photon-counting measurement, the Cramér-Rao bound links the FI to the best achievable precision of any unbiased phase estimator, $\Delta\phi^2 \geq 1/[\nu F_{tot}(\phi)]$, where $\nu$ is the number of measurement repetitions[33]. Consequently, the peak value of $F_{tot}(\phi)$ sets the maximum phase sensitivity attainable with our whole interferometric configuration. Figure 4c, f plots the maximum FI as a function of $|A|^2$. In both configurations, $F_{|\psi_i\rangle}^{max}$ peaks in the ancilla port at $|A|^2 = 0.5$ (0.81 for type 1, 0.83 for type 2), and the total FI $F_{tot}(\phi)$ reaches unity in both cases (see Supplementary Figs. 5c, d, and 6c, d). For type 1, the fields interfering at MZI$_3$ have a fixed zero relative phase, so reducing $|A|^2$ only rescales the relative amplitudes reaching MZI$_3$, leading to a uniform reduction of interference visibility and hence a simultaneous suppression of FI across all output modes. In contrast, for type 2, the fields interfere in fixed quadrature, so changes in $|A|^2$ modify how the input phase-dependent amplitudes are redistributed at MZI$_3$, causing phase sensitivity to shift from the ancilla to the signal ports as $|A|^2$ is reduced. This behavior illustrates the circuit's capability for tunable phase-encoded signal routing and redistribution of phase sensitivity among the output modes (see Supplementary Note 4 (Subsection 4.4)).

# Beam splitter with π-shifted reflection (Type1) - NOON state experiment

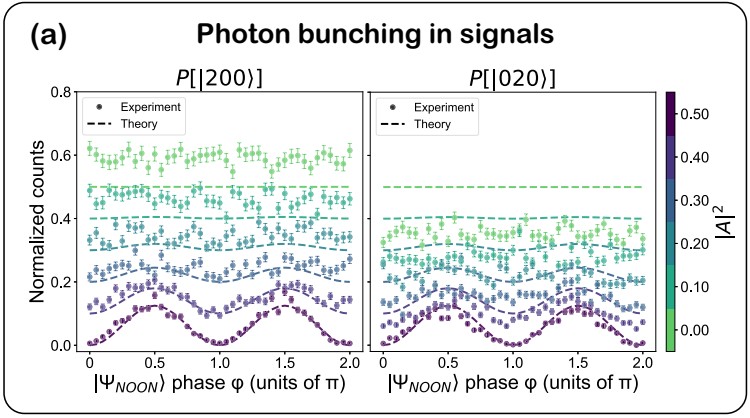

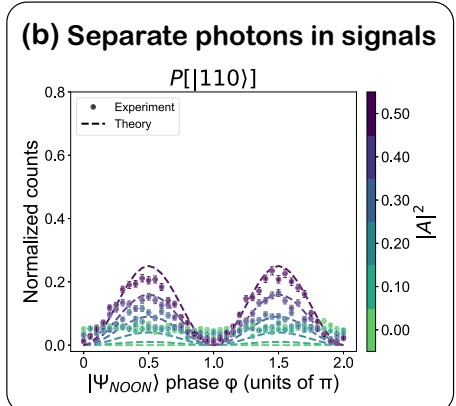

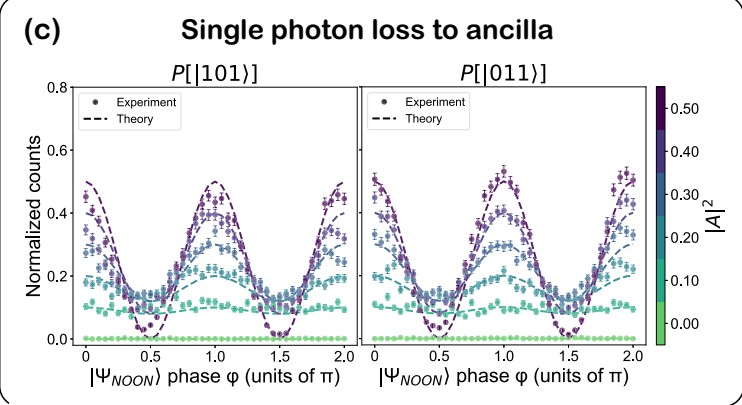

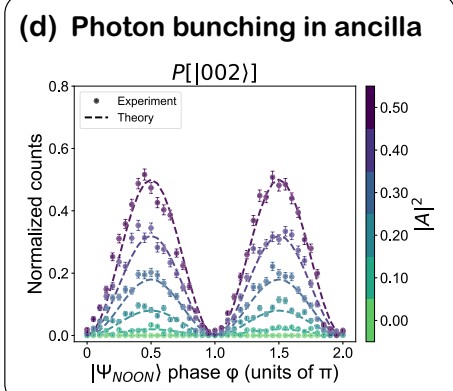

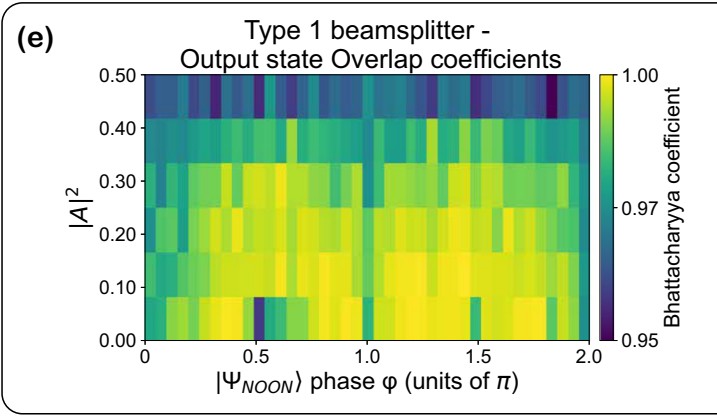

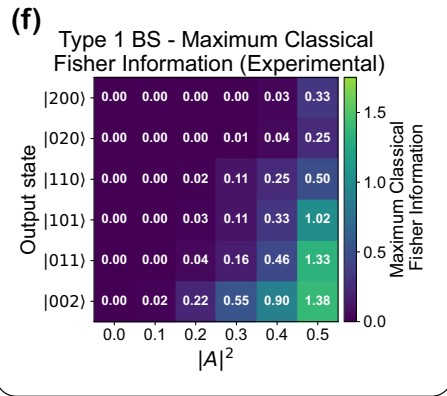

**Fig. 5 | Beam splitter with π-shifted reflection (type 1)-NOON state experiment.**
**a–d** Experimentally measured and normalized counts, and theoretically predicted probabilities for various output Fock states, plotted as a function of the input state phase $\phi$, with $|A|^2$ represented by the color scale. Probabilities are normalized to the total coincidence counts across all output states and corrected for variations in detector efficiency and coupling efficiency of the chip's output waveguides to the corresponding fibers. The lines denote theoretical predictions; symbols are experimental data. The theoretical data are obtained by applying the CPA unitary transformation matrix $S_{total}$ to the input state vector and computing output

intensities. For type-1, the MZI$_3$ input phase difference is fixed to $\Delta\phi_{in} = 0$ (see Eq. 3 and Supplementary Note 4 (Subsection 4.4)). Error bars represent one standard deviation, calculated as $\sqrt{N}$ from Poisson counting statistics, with proper propagation through the normalization procedure. **e** Bhattacharyya overlap coefficients between experimental and theoretical distributions across $\phi$ and $|A|^2$, evaluated in the Fock basis, demonstrating high-fidelity implementation of the beam splitter transformations. **f** Maximum classical Fisher information extracted for each output Fock state at different $|A|^2$ values (See Eq. 6).

## NOON state $\left(|\Psi_{NOON}\rangle\right)$ experiment

Figure 3c shows the Hong-Ou-Mandel interference curve measured at the state-preparation MZI used to generate the NOON state. The dip, with a visibility of 93.9%, confirms high photon indistinguishability, temporal overlap, and state purity. Spectral overlap data for the photon pair (at 1569.5 nm) is provided in "Methods" (section "The SPDC source").

Figures 5 and 6 present results for the NOON state experiment with type 1 and type 2 beam splitter configurations, respectively. Coincidence counts from all 15 detector pairs are processed to extract photon statistics for the six two-photon Fock basis states: $|200\rangle$, $|020\rangle$, $|110\rangle$, $|101\rangle$, $|011\rangle$, and $|002\rangle$, where the ket is ordered as $|S_1, S_2, \text{Anc}\rangle$.

In the NOON-state case, coherent absorption is governed by programmable two-photon interferences at MZI$_1$ and MZI$_3$, together

## Symmetric (Type 2) beamsplitter NOON state experiment

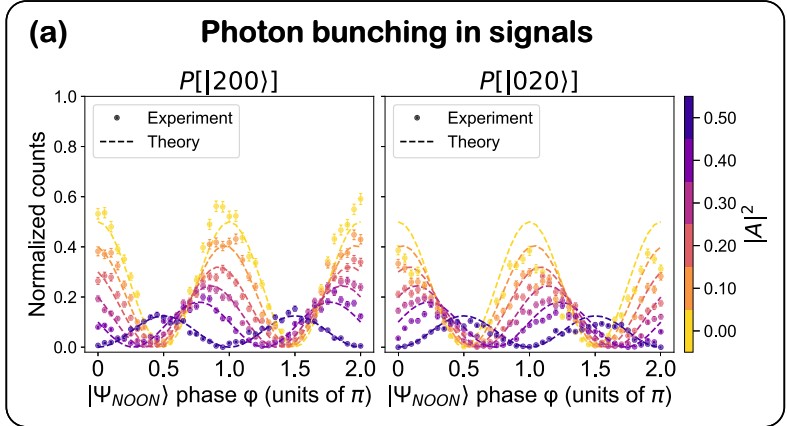

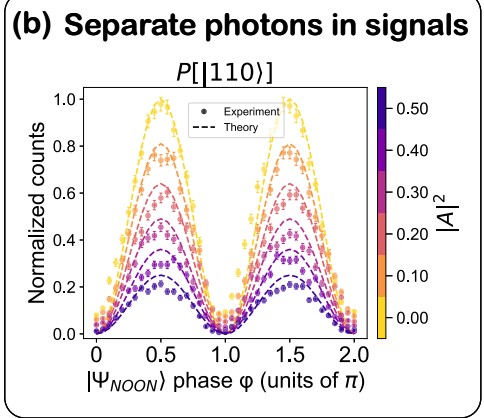

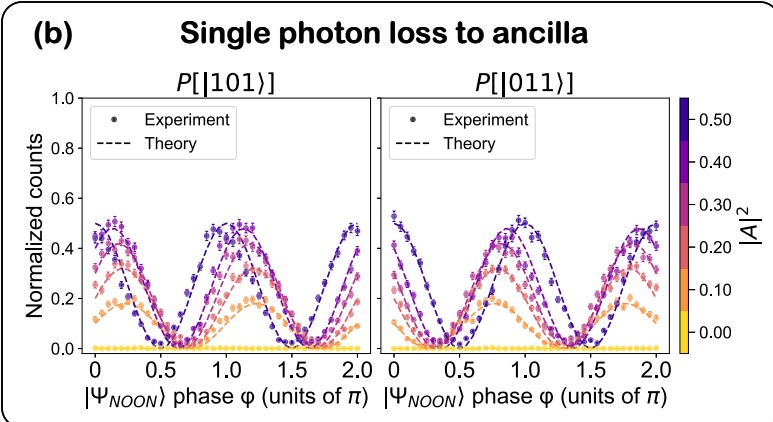

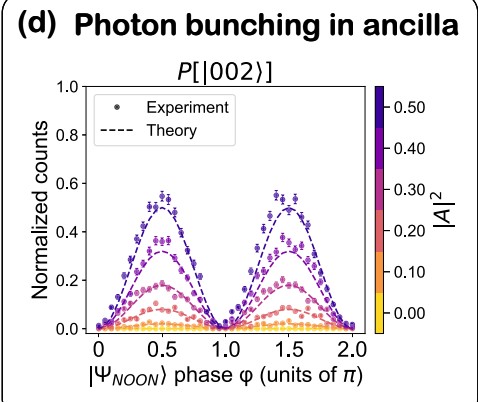

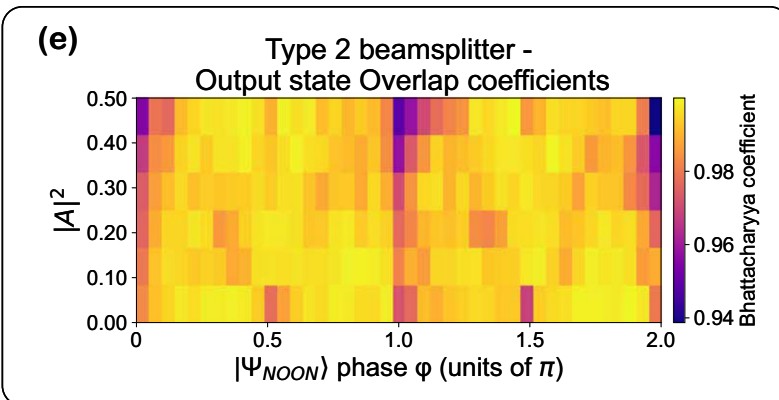

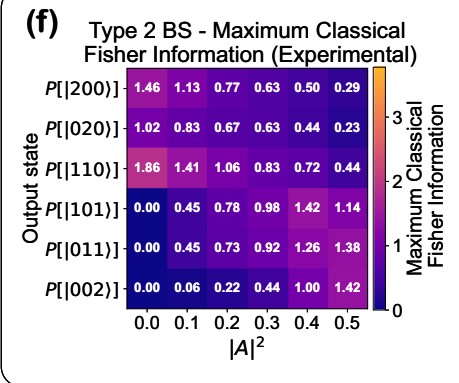

**Fig. 6 | Symmetric beam splitter (type 2)-NOON state experiment.**
**a–d** Experimentally measured and normalized counts, and theoretically predicted probabilities for various output Fock states, plotted as a function of input state phase $\phi$, with $|A|^2$ represented by the color scale. Probabilities are normalized to the total coincidence counts across all output states and corrected for variations in detector efficiency and coupling efficiency of the chip's output waveguides to the corresponding fibers. The lines denote theoretical predictions; symbols are experimental data. The theoretical data are obtained by applying the CPA unitary transformation matrix $S_{total}$ to the input state vector and computing output

intensities. For type-2, the MZI$_3$ input phase difference is fixed to $\Delta\phi_{in} = \pi/2$ (see Eq. 3 and Supplementary Note 4 (Subsection 4.4)). Error bars represent one standard deviation, calculated as $\sqrt{N}$ from Poisson counting statistics, with proper propagation through the normalization procedure. **e** Bhattacharyya overlap coefficients between experiment and theory across $\phi$ and $|A|^2$, evaluated in the Fock basis, demonstrating high-accuracy implementation of the beam splitter transformations. **f** Maximum classical Fisher information extracted for each output Fock state at different $|A|^2$ values (See Eq. 6).

with controllable single- and two-photon coupling to the ancilla at MZI$_2$. After MZI$_1$, the input state $|\Psi_{NOON}\rangle$ is transformed onto a coherent superposition of $|2_+\rangle = (|20\rangle + |02\rangle)/\sqrt{2}$ and $|11\rangle$ states, whose amplitudes vary as $\sin\phi$ and $\cos\phi$, respectively. For in-phase inputs ($\phi = 0, 2\pi,...$), the state populates $|2_+\rangle$, and for out-of-phase inputs ($\phi = \pi, 3\pi,...$), it populates $|11\rangle$, and for intermediate phases, both contribute. The portion of light emerging in the lower arm after MZI$_1$ couples to the ancilla at MZI$_2$, while the remaining amplitudes interfere again at MZI$_3$ (this quantum interference is dictated also by the phase $\phi_{rt}$) to produce the final output distributions. This framework captures the discrete-variable Fock states coherent absorption process described by Vetlugin et al.[23], extended here to programmable

interferometric control of multiphoton absorption. A more detailed theoretical framework of the quantum state evolution in the circuit is provided in Supplementary Note 4 (Subsection 4.2). Operating in the quantum regime, these experiments reveal hallmark signatures of genuine multiphoton interference: unlike the classical-like $2\pi$-periodic fringes for single photons, the $\pi$-periodic oscillations here directly reflect the $e^{i2\phi}$ phase dependence and two-photon path entanglement intrinsic to NOON states (Eq. 5).

In the type 1 (fixed $\phi_{rt} = \pi$) configuration, the output probabilities of all six Fock basis states exhibit strong phase sensitivity at high $|A|^2$ values. As $|A|^2$ decreases, this phase sensitivity diminishes across all Fock state components, resulting in flatter modulation curves and reduced visibility in the interference patterns. As predicted by theory, at maximal $|A|^2$ and phases $\phi = 0$, $\pi$, and $2\pi$, the system exhibits deterministic single-photon absorption: exactly one photon is routed to the ancilla with near-unity probability, while the remaining photon emerges in either of the two signal modes with equal probability. At the intermediate phases $\phi = 0.5\pi$ and $1.5\pi$, the behavior shifts to probabilistic two-photon absorption, where both photons are absorbed with 50% probability, and the bunching probability of the absorbed photons approaches 100%.

The experimental data shown in Fig. 5a–d reflect these trends, highlighting coherent phase-dependent control over quantum interference pathways. Additionally, the probability of photon bunching in the signal modes generally decreases with increasing $|A|^2$, whereas the probability of photon antibunching increases. At intermediate values of $\phi$ and $|A|^2$, the system yields a mixture of all Fock state components, consistent with theoretical predictions for general lossy beam splitters.

In the case of the $|200\rangle$ and $|020\rangle$ states, deviations from the ideal curves are observed. These deviations are consistent with a small residual asymmetry in the effective two-port transformation implemented by $MZI_3$. This asymmetry becomes apparent when the effective input to $MZI_3$, after propagation through the preceding stages of the circuit, is a symmetric two-photon state (e.g., $|11\rangle$ or the symmetric $NOON^+$ state). Plausible physical origins of this asymmetry include fabrication-induced imbalance of the MMI couplers, differential propagation loss between the two MZI arms, and weak parasitic reflections or mode mismatch at waveguide junctions, all of which can break the ideal port symmetry of $MZI_3$ while remaining largely invisible to first-order transmission measurements. All other phase shifters and MZIs in the programmable circuit operate as expected. A detailed quantitative analysis supporting this interpretation is provided in Supplementary Note 6.

While very slight shifts from the ideal theory lines are present for some other output Fock state curves also (probably due to cumulative minor imperfections in the overall experimental setup), the overall similarity between the measured and predicted output Fock state probability distributions, quantified by the Bhattacharyya coefficient (Figs. 5e and 6e) (discussed later), remains high, and the overall trends with varying $|A|^2$ and input phase are consistent with theoretical predictions.

In the type 2 (symmetric $|t| = |r|$) configuration, deterministic single-photon absorption occurs at $|A|^2 = 0.5$ and $\phi = 0$, $\pi$, $2\pi$, while probabilistic two-photon absorption with perfect bunching in the ancilla occurs at $\phi = 0.5\pi$ and $1.5\pi$-mirroring the behavior observed in type 1. However, several key differences emerge. Unlike type 1, where the phase-dependent modulation of all Fock state probabilities flattens as $|A|^2 \to 0$, the type 2 configuration retains significant visibility in several output components even at low absorption. Only the curves of $P(|101\rangle)$, $P(|011\rangle)$, and $P(|002\rangle)$ flatten at $|A|^2 = 0$, while the two photon signal terms $(P(|200\rangle)$, $P(|020\rangle)$, $P(|110\rangle))$ terms remain phase-sensitive.

At low absorption and $\phi = 0$, $\pi$, the output exhibits deterministic bunching in the signal ports and at $\phi = 0.5\pi$, $1.5\pi$, it switches to

deterministic antibunching or anti-coalescence, highlighting the programmable interference nature of the symmetric configuration. Here, anti-coalescence refers to the phase-controlled suppression of same-mode two-photon output, accompanied by enhanced coincidence across distinct output modes. A distinguishing feature is the continuous phase shift of the oscillation curves for $P(|200\rangle)$, $P(|020\rangle)$, $P(|110\rangle)$, and $P(|011\rangle)$ as the absorption is varied. A full shift of $\pi/2$-half the fringe period-is observed between minimum and maximum absorption values, indicating a dynamic change in the interference landscape. Finally, in contrast to type 1, both signal bunching and antibunching probabilities decrease with increasing absorption. Figures 5e and 6e show the Bhattacharyya overlaps between experimentally measured and theoretically predicted output distributions across the six-dimensional Fock basis. The Bhattacharyya coefficient is defined as

$$B(P, Q) = \sum_{i=1}^{6} \sqrt{p_i q_i}, \tag{8}$$

where $P = \{p_i\}$ and $Q = \{q_i\}$ are the normalized experimental and theoretical probabilities, and the index $i$ runs over the six-dimensional two-photon Fock basis[35]. A value of $B = 1$ indicates perfect agreement. Across all absorption and phase settings, we observe overlaps exceeding 0.93, with average values of $0.9850 \pm 0.0122$ for the type 1 configuration and $0.9910 \pm 0.0092$ for the type 2 configuration, demonstrating high-fidelity implementation of the target transformations. The residual deviation from unity is primarily attributed to small deviations of the $|200\rangle$ and $|020\rangle$ Fock-state probabilities from their ideal values, arising from residual port asymmetry of the effective two-port transformation (as discussed earlier), with all other experimental imperfections contributing only minor corrections. These include polarization-dependent effects in the MMI couplers and SNSPDs, slow polarization drifts in single-mode fibers, and small thermal or phase fluctuations during operation. The maximum classical FI $F_{|\psi_i\rangle}^{\max}$ (Figs. 5f and 6f), calculated using Eq. 6, captures the sensitivity of each output mode to input phase changes and the total classical FI, $F_{tot}(\phi)$, calculated using Eq. 7, captures the total phase sensitivity of the circuit to the input state with phase $\phi$. Similar to the single-photon experiment, the type 1 beam splitters exhibit peaks in classical FI at maximum absorption, particularly in the Fock states with one or two photons in the ancilla ($|002\rangle$, $|101\rangle$, $|011\rangle$). As $|A|^2$ decreases, the FI across all Fock states diminishes. In contrast, the type 2 configuration enables redistribution of phase sensitivity among different Fock components through absorptivity tuning. All Fisher information values reported in this work were extracted from sinusoidal fits to the measured intensity modulations; the fit curves and theoretical predictions are presented in Supplementary Note 5 (See Supplementary Figs. 5–8).

The maximum measured total FI, $F_{tot}(\phi)$, reaches 3.41 for the type 1 configuration and 3.42 for the type 2 configuration (see Supplementary Figs. 7 and 8c, d), surpassing the shot-noise limit $F_{SQL} = N = 2$ for two photons and approaching the Heisenberg limit $F_{HL} = N^2 = 4$, which represents the ultimate quantum precision bound for phase estimation with $N = 2$ particles[36].

For the NOON state case also, the circuit can be used to distinguish between orthogonal superposition states. In both configurations, the antisymmetric NOON state $|2_-\rangle = \frac{1}{\sqrt{2}}(|20\rangle - |02\rangle)$ (corresponding to $\phi = 0, \pi, 2\pi, ...$) leads to deterministic single-photon loss into the ancilla. In contrast, the symmetric NOON state $-|2_+\rangle = -\frac{1}{\sqrt{2}}(|20\rangle + |02\rangle)$ (for $\phi = \frac{\pi}{2}, \frac{3\pi}{2}, ...$) results in both photons being detected either together in the signal outputs or both directed into the ancilla. As mentioned in an earlier section, the choice of the specific input state that leads to deterministic single-photon absorption can be coherently tuned by adjusting the phase $\phi_{MZI_1}$.

## Discussion

Our experimental results establish a framework for simulating non-unitary quantum transformations using programmable linear optics on an integrated photonic chip. By combining quantum state preparation, programmable circuitry, and photon number-resolving detection, we emulate and characterize coherent absorption of quantum light in a fully tunable, ancilla-assisted linear optical architecture.

The two input states-single photon and two photon NOON states, probe distinct interference regimes. The single photon state shows classical coherent absorption-like oscillations with $2\pi$ periodicity, while the NOON state exhibits $\pi$ periodic modulation and enhanced phase sensitivity, highlighting the metrological advantage of multiphoton states[36]. The peak total classical FI reaches $F_{tot}(\phi) = 1$ for single photon states and $F_{tot}(\phi) = 3.4$ for NOON states, the latter surpassing the shot noise limit ($F = 2$) and approaching the Heisenberg limit ($F = 4$) for two photons.

Comparing the two beam splitter configurations, type 1 (with fixed $\phi_{rt} = \pi$) imposes a static interference symmetry, while type 2 (symmetric, $|t| = |r|$) allows dynamic modulation of the internal phase $\phi_{rt}$ through the intrinsic absorptivity. This tunability enables programmable redistribution of phase sensitivity across output modes. In particular, type 2 facilitates controlled distribution of classical FI among Fock basis components. As a result, each detection outcome carries tailored phase sensitivity, enabling state-selective readout, adaptive quantum sensing, and multiplexed metrology schemes, where phase information is actively steered and shaped within the optical circuit[37].

By transforming a two-mode, two-photon input into a programmable three-mode Fock state distribution, our circuit provides a scalable platform for Fock state engineering and quantum state filtering. Its programmable loss-emulating design is suitable for simulating dissipative channels in quantum thermodynamics and non-Markovian open system dynamics, as well as realizing arbitrary POVM in an enlarged Hilbert space[1,5]. The three-mode non-unitary block demonstrated here can serve as a reconfigurable subunit in larger photonic quantum processors, enabling more complex functionalities within modular photonic networks. As loss is emulated via an ancilla mode, no photon is destroyed, but rerouted, enabling its independent measurement or reinjection into the main circuit block in a later stage if desired. These capabilities expand the functional repertoire of linear optical systems and open new possibilities for non-unitary quantum photonic protocols.

## Methods

### Lossy beam splitter constraints

To ensure physical validity, any lossy beam splitter must preserve the canonical bosonic commutation relations, such as $[\hat{a}_{out}, \hat{a}_{out}^{\dagger}] = 1$ and $[\hat{a}_{out}, \hat{b}_{out}^{\dagger}] = 0$. For a port-symmetric beam splitter with equal coupling to the loss channel, these constraints lead to an additional phase-dependent relation between $t$, $r$, and $A$:

$$2|t||r| \cos \phi_{rt} = \pm |A|^2. \qquad (9)$$

Together with the energy conservation identity

$$|A|^2 = 1 - |t|^2 - |r|^2, \qquad (10)$$

These equations place a fundamental upper bound on the intrinsic absorption coefficient:

$$|A|^2 \le 2|t||r| \le |t|^2 + |r|^2 = 1 - |A|^2, \qquad (11)$$

implying that for a lossy beam splitter, the maximum possible intrinsic absorption coefficient is $|A|^2 = 0.5$.

The procedure used to determine the complex transmission and reflection coefficients t and r from a specified absorption value for both beam splitter types is detailed in Supplementary Note 1 (Subsection 1.1).

### Quasiunitary decomposition scheme

To implement a non-unitary transformation using linear optical elements, we employ the quasi-unitary extension scheme developed in ref. [11]. A given non-unitary $2 \times 2$ scattering matrix S representing a lossy beam splitter is first decomposed via singular value decomposition (SVD):

$$T = UDW, \qquad (12)$$

where U and W are $2 \times 2$ unitary matrices, and D is a diagonal matrix of singular values $0 < \sigma_i \le 1$. Each $\sigma_i < 1$ corresponds to a lossy mode and requires the introduction of an ancilla mode. The total enlarged system has dimension $N = 2 + N_{ancilla}$, and the corresponding unitary embedding is constructed as

$$S_{total} = S_U \cdot S_D \cdot S_W, \qquad (13)$$

where $S_U$ and $S_W$ are padded versions of U and W acting on the full N-mode space, and $S_D$ implements beam splitter-like couplings between lossy signal modes and ancillary vacuum inputs, with attenuation strengths determined by $\sigma_i$.

This construction was implemented numerically using custom Python code that automatically determines the required number of ancilla modes and constructs the enlarged unitary $S_{total}$ for arbitrary $M \times M$ non-unitary transformations (with M any positive integer, $M = 2$ in our case).

For all beam splitter configurations simulated in this work, the target transformations belong to the class of reciprocal, port-symmetric lossy beam splitters with a single effective loss channel. As a consequence, the associated non-unitary $2 \times 2$ transfer matrix T satisfies rank$(I - T^{\dagger}T) = 1$, i.e., exactly one singular value of T differs from unity (see Supplementary Note 1 (Subsection 1.2)). Within the quasi-unitary dilation framework, this implies that a single ancilla mode is both necessary and sufficient to embed the lossy transformation into a larger unitary. Accordingly, all losses in the simulated devices can be consistently modeled using one ancilla mode, and the effective ancilla-embedded transformation is a $3 \times 3$ unitary acting on the two signal modes and one ancilla mode.

More generally, for an $m \times m$ passive non-unitary transformation $T_{net}$ (e.g., the net map of a larger lossy network), the minimal number of ancilla modes is $m_{anc} = \mathrm{rank}\left(I - T_{net}^{\dagger}T_{net}\right)$, so the enlarged unitary acts on $N = m + m_{anc}$ modes. A universal Clements mesh on N modes requires $N_{MZI} = N(N-1)/2$ MZIs (and $\approx 2N_{MZI}$ thermo-optic phase shifters), with the additional overhead specific to programmable loss set by $m_{anc}$ rather than by the number of lossy $2 \times 2$ sub-blocks.

### Clements decomposition scheme

The enlarged $3 \times 3$ unitary $S_{total}$ is decomposed into a mesh of two-mode unitary operations using the Clements decomposition[3]. Each $2 \times 2$ unitary is implemented with an MZI, composed of two balanced 50:50 beam splitters and a pair of phase shifters: an internal phase $\theta$ and an external phase $\phi$.

The transfer matrix of a single MZI is expressed as:

$$U_{MZI}(\theta, \phi) = e^{i\left(\frac{\theta}{2} + \frac{\pi}{2}\right)} \begin{pmatrix} e^{i\phi} \sin\left(\frac{\theta}{2}\right) & \cos\left(\frac{\theta}{2}\right) \\ e^{i\phi} \cos\left(\frac{\theta}{2}\right) & -\sin\left(\frac{\theta}{2}\right) \end{pmatrix}. \qquad (14)$$

These MZIs are arranged in a rectangular mesh acting on adjacent modes to synthesize the full unitary transformation. We used a self-modified version of the open-source `interferometer` Python

package (v1.1.1, installed via `pip` and then modified; https://pypi.org/project/interferometer/) to implement the decomposition, adapting its conventions to match our device's MZI transfer matrix. For the CPA transformations used in this work, the Clements mesh contains three MZIs arranged in two layers.

## The photonic chip, thermal stabilization and control

The photonic circuit is implemented on a silicon-on-insulator platform, featuring silicon waveguides with a height of 250 nm and a width of 450 nm, fabricated atop a 3 μm buried oxide layer. The waveguides are designed to support the fundamental TE mode light at around 1550 nm. Thermo-optic phase shifters embedded on top of the waveguides provide reconfigurable control over the internal ($\theta$) and external ($\phi$) phase shifts of each MZI. The two beam splitters forming each MZI are balanced 2 × 2 multimode interference (MMI) couplers.

As illustrated in Fig. 3, the photonic chip is mounted on a Peltier-cooled stage to suppress thermal drift. The chip is also packaged to ensure stable optical coupling from and to the fiber arrays, in addition to thermal stability. Electrical currents for driving the phase shifters are delivered via wire bonds connected to a custom printed circuit board (PCB) assembly. Each phase shifter is driven using a low-noise current source with PID feedback to ensure precise phase stability.

The SPDC crystal is similarly mounted on a temperature-controlled stage, enabling fine control of its phase-matching condition to generate wavelength-degenerate photon pairs. This ensures high photon indistinguishability−critical for the fidelity of the two-photon NOON state. Before each measurement run, the chip is calibrated using classical light to determine the current-phase mapping of each MZI, ensuring accurate phase programming. For additional details about the experimental setup, refer to Supplementary Note 3.

## Detection and data acquisition

Single-photon coincidence counts were measured using superconducting nanowire single-photon detectors (SNSPDs) manufactured by single quantum B.V. All detectors exhibited system detection efficiencies exceeding 70%, dark count rates below 10 Hz, and timing jitter below 20 ps. Detector dead times were negligible on the timescales relevant to the present measurements. Photon arrival times were recorded using a multi-channel time-tagging module, enabling full reconstruction of coincidence events and photon-number-resolved statistics via post-processing.

To account for unequal detection efficiencies across channels, we performed a calibration procedure in which light was sequentially routed to each output channel while keeping all other experimental conditions unchanged. The resulting single-photon count rates were used to extract relative detection efficiencies, which were subsequently employed to normalize the measured coincidence probabilities. This procedure corrects for channel-dependent losses arising from detector efficiency variations, fiber coupling, and passive optical components.

## The SPDC source

Photon pairs were generated via SPDC in a nonlinear crystal pumped by a continuous-wave diode laser (Cobolt), operated at a drive current of 220 mA. The laser emitted at a central wavelength of 785 nm with an output power of 70 mW at the crystal input. When characterized directly at the output of the SPDC source, prior to injection into the photonic circuit, we measured single-photon detection rates of approximately 150 kHz in each channel. The corresponding two-photon coincidence rate between the signal and idler channels was approximately 30 kHz at zero relative delay, evaluated using a coincidence window of 2 ns, corresponding to a heralding efficiency of about 20%.

To confirm the single-photon nature of our source, we performed a heralded second-order correlation measurement using a standard

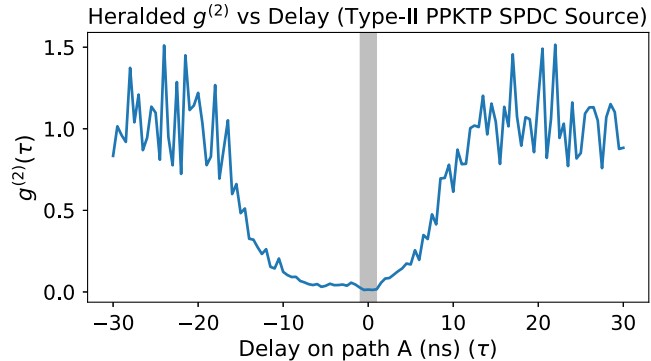

**Fig. 7 | Heralded $g^{(2)}$ measurement.** Second-order correlation function $g^{(2)}(\tau)$ of output 1 from the heralded single-photon source. The shaded grey region around zero delay indicates the 2 ns coincidence window used for all time-correlated photon counting measurements throughout this work.

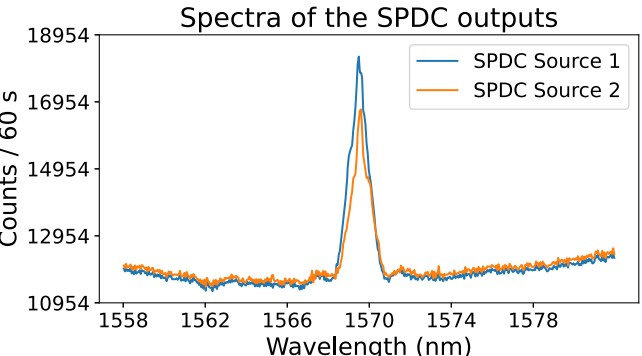

**Fig. 8 | Spectral overlap of the SPDC sources.** Measured emission spectra of the two SPDC source outputs used in the experiments, recorded under identical pump and collection conditions. The high degree of spectral overlap ensures the indistinguishability of the photons, which is essential for generating high-fidelity NOON states and the Hong-Ou-Mandel dip. Out 1 peak is centered at 1569.48 nm and out 2 peak at 1569.59 nm. This overlap was achieved by tuning the temperature of the temperature control stage to 28.5 °C.

three-detector Hanbury Brown-Twiss configuration (Fig. 7). The light from output 1 of the type-II PPKTP SPDC source was split using a 50:50 fiber beamsplitter and detected at two outputs, labeled A and B, while the idler photon was used as a heralding trigger. The second-order correlation function $g^{(2)}(\tau)$ was computed from the three-fold coincidence counts among all detectors.

The heralded second-order correlation function is given by

$$g^{(2)}(\tau) = \frac{R_{ABH}(\tau)\, R_H}{R_{AH}\, R_{BH}(\tau)}, \tag{15}$$

where $R_{ABH}(\tau)$ is the rate of three-fold coincidences between detectors $A$, $B$, and the heralding detector $H$ at time delay $\tau$, $R_{AH}$ and $R_{BH}(\tau)$ are the two-fold coincidence rates between detectors $A$ and $H$, and $B$ and $H$ respectively, and $R_H$ is the single count rate of the herald. A dip in $g^{(2)}(\tau)$ around $\tau = 0$ is a signature of a single-photon state.

We measured an excellent $g^{(2)}(0) = 0.014$, indicating strong suppression of multi-photon contributions and validating the use of this source for single-photon experiments. The measurement shown in Fig. 7 corresponds to output 1 of the SPDC source, which was used in all single-photon experiments. The maximum value of $g^{(2)}(\tau)$ within the 2 ns coincidence window highlighted by the shaded grey area in the figure is 0.0273. Figure 8 shows the spectral overlap of the photon pairs obtained by tuning the temperature of the SPDC crystal to reach degeneracy, thereby producing indistinguishable photons.

## Coherent absorption theoretical calculations

Theoretical phase-dependent intensity curves at various $|A|^2$ values are calculated by applying the corresponding lossy beam splitter scattering matrix to the input state vector via standard matrix multiplication, as defined in Eq. 16 for the single-photon experiment.

$$\begin{pmatrix} \sqrt{P_{S_1}} \\ \sqrt{P_{S_2}} \\ \sqrt{P_{\text{Anc}}} \end{pmatrix} = \begin{pmatrix} t & r & R_{13} \\ r & t & R_{23} \\ R_{31} & R_{32} & R_{33} \end{pmatrix} \cdot \frac{1}{\sqrt{2}} \begin{pmatrix} e^{i\phi} \\ -1 \\ 0 \end{pmatrix}, \quad (16)$$

where the transformation matrix is the enlarged matrix resulting from the quai-unitary extension scheme (see Fig. 2a), $\phi$ is the input single photon quantum state phase, and $P_{S_1}$, $P_{S_2}$, and $P_{\text{Anc}}$ are the probabilities of corresponding to signal 1, signal 2, and Ancilla, respectively. For the NOON-state predictions, we use the open-source qoptcraft Python package (v2.0.0, installed via pip; https://pablovegan.github.io/QOptCraft/#quantizing-linear-interferometers), which maps the $3 \times 3$ CPA scattering matrix to a $6 \times 6$ unitary transformation in the two-photon, three-mode Fock basis, following the generalized bosonic scattering theorem—also known as the permanent rule. This method is implemented in qoptcraft as described in ref. 38. For clarity, this mapping is illustrated in Eq. 17.

$$\begin{pmatrix} t & r & R_{13} \\ r & t & R_{23} \\ R_{31} & R_{32} & R_{33} \end{pmatrix} \xrightarrow[\text{Fock basis conversion}]{\text{QOptCraft}} \begin{pmatrix} F_{11} & F_{12} & \cdots & F_{16} \\ F_{21} & F_{22} & \cdots & F_{26} \\ \vdots & \vdots & \ddots & \vdots \\ F_{61} & F_{62} & \cdots & F_{66} \end{pmatrix}.$$

Single $-$ photon, 3 mode
$3 \times 3$ scattering matrix

Two $-$ photon, 3 mode fock basis
$6 \times 6$ unitary transformation matrix

$$(17)$$

Following the above mapping, the predicted output probability amplitudes for the NOON-state experiment are obtained by multiplying the $6 \times 6$ unitary with the input state vector in the two-photon, three-mode basis:

$$\begin{pmatrix} \sqrt{P_{|200\rangle}} \\ \sqrt{P_{|020\rangle}} \\ \sqrt{P_{|110\rangle}} \\ \sqrt{P_{|101\rangle}} \\ \sqrt{P_{|011\rangle}} \\ \sqrt{P_{|002\rangle}} \end{pmatrix} = \begin{pmatrix} F_{11} & F_{12} & \cdots & F_{16} \\ F_{21} & F_{22} & \cdots & F_{26} \\ \vdots & \vdots & \ddots & \vdots \\ F_{61} & F_{62} & \cdots & F_{66} \end{pmatrix} \cdot \frac{1}{\sqrt{2}} \begin{pmatrix} e^{i2\phi} & \leftarrow |200\rangle \\ 0 & \leftarrow |110\rangle \\ 0 & \leftarrow |101\rangle \\ -1 & \leftarrow |020\rangle \\ 0 & \leftarrow |011\rangle \\ 0 & \leftarrow |002\rangle \end{pmatrix}, \quad (18)$$

where each $P_{|\cdot\rangle}$ corresponds to the probability of detecting photons in the mode occupation indicated by the ket label (S1, S2, and Anc).

## Data availability

The raw data and processing codes corresponding to this study are openly available at https://zenodo.org/records/19596162.

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

## Acknowledgements

The authors thank Prof. Ivan M. Khaymovich for helpful discussion. G.K., A.W.E. acknowledges the support from Knut and Alice Wallenberg (KAW) Foundation through the Wallenberg Center for Quantum Technology (WACQT). J.G. acknowledges support from the Swedish Research Council (ref: 2023-06671 and 2023-05288), Vinnova project (ref: 2024-00466) and the Göran Gustafsson Foundation. A.W.E. acknowledges support from the Swedish Research Council (VR) Starting Grant (ref: 2016-03905), and Vinnova quantum kick-start project 2021. V.Z. acknowledges support from the KAW.

## Author contributions

G.K. conceived the project together with J.G., A.W.E., G.K. and R.Y. developed the chip control, calibration software, and control electronics; G.K. developed the multichannel coincidence counting software and transformation matrix generation codes. The SPDC source setup was originally constructed by J.G. G.K. optimized the SPDC source and set up the optical measurement apparatus with contributions from S.O. G.K. led all optical measurements with contributions from J.G., S.O., and performed data analysis, visualization, and manuscript preparation with input from all authors. A set of 8 × 8 photonic chips was packaged by V.D., S.P. The project was supervised by A.W.E., J.G. and V.Z. Funding for the project was secured by J.G., A.W.E. and V.Z.

## Funding

## Competing interests

The authors declare no competing interests.

## Additional information

**Supplementary information** The online version contains Supplementary material available at https://doi.org/10.1038/s41467-026-72850-6.

