## [Transparent Peer Review File · Nature Communications]

Emulation of Coherent Absorption of Fock-State Quantum Light in a Programmable Linear Photonic Circuit

Corresponding Author: Professor Ali Elshaari

Version 0:

Reviewer comments:

Reviewer #1

(Remarks to the Author)

Manuscript report:

Title: Emulation of Coherent Absorption of Quantum Light in a Programmable Linear Photonic Circuit

In this manuscript, the authors report on an experimental effort to demonstrate nonunitary transformations to emulate coherent absorption of quantum light. Their approach is based on a programmable linear photonic platform where loss is implemented through ancilla modes.

The authors demonstrate the well-established concept of coherent perfect absorption (CPA) using a versatile programmable photonic circuit. Their platform can tune the all-physical parameters to implement PCA in a broad range of single-photon and two-photon quantum states. This is a very appealing work, and I am positive in recommending this paper for publication in Nature Communications, provided the comments detailed below are addressed.

Suggestions for improving the presentation of the paper.

In general, the authors present very convincing results unequivocally demonstrating the effects of PCA. However, I find the writing of the manuscript quite technical, making the concepts and the description of the device operations hard to follow. As it is written, the paper would perfectly fit for a more specialized Journal.

For instance, the authors did a good job describing how coherent absorption arises in classical systems. Specifically they State:

“Classically, coherent absorption arises when counterpropagating waves form a standing wave, with the absorption determined by the absorber’s position within the resulting field profile. Complete absorption or coherent perfect absorption (CPA) occurs at antinodes, full transmission at nodes, and partial absorption elsewhere.”

This statement is crucial and clearly explains the phenomenon of classical coherent absorption. I suggest the authors to elaborate the concepts for the quantum cases in a similar fashion and perhaps accompanying such explanations with a couple of figures visually illustrating the physical processes.

In the present form, the manuscript just describes the experimental platform, the quantum states used and the results. That is, it lacks a physical description of the effects and how specifically the states evolve through the system.

Additionally, why the authors decided to use the Clemence configuration?

Why the Fisher information is a good estimator to quantify the phase sensitivity of the device? The authors must explain in a few words the physical meaning of Eq. (6), not just state what it does?

The results presented in the manuscript are compelling, but the overall clarity and structure need to be improved before the work can be considered for Nature Communications.

(Remarks on code availability)

Reviewer #2

(Remarks to the Author)

This manuscript demonstrates a programmable, on-chip emulation of coherent absorption (CPA) by embedding a lossy 2×2 transformation within a larger unitary via a single ancilla mode and providing closed-form settings that map absorptivity and internal phase directly to MZI controls. The experiments with single- and two-photon probes reproduce the predicted phase dependences and output distributions with high overlap, and the reported Fisher-information enhancement (approaching the $N=2$ Heisenberg limit) substantiates the metrological claims. Beyond prior fixed CPA implementations, the work elevates CPA to a programmable primitive that is firmly grounded in quasi-unitary dilation and Clements-mesh synthesis, and should be impactful for programmable non-unitary photonics, POVM engineering, and multiplexed quantum sensing. The methodology is sound and meets community standards, the conclusions are supported by the data. I see no publication-blocking flaws, only minor clarifications that would strengthen transparency: a brief on-chip calibration validating the closed-form settings, a compact error budget for the residual distribution mismatches (including MZI imbalance and detector effects), and basic detector specifications alongside per-detected-photon FI. Reproducibility is good at the level of theory and procedures; to make it turnkey for others, I recommend sharing representative raw time-tag histograms with analysis scripts and a short, version-pinned code/README to regenerate key plots. I find the work technically solid, original in its programmable framing, and suitable for publication after minor revision.

General assessment: Accept after minor revision.

Rationale: The results are robust and well-presented. I suggest the authors address the points below to tighten validation, expand clarity on limits and scalability, and enhance reproducibility. None appear to require substantial new experiments beyond calibrations/plots and modest analysis.

Major comments

1. Conceptual scope of the non-unitary emulator. Please articulate conditions under which a single ancilla suffices for port-symmetric lossy beam splitters and when additional ancilla modes become necessary for more general non-unitary maps (e.g., asymmetric loss, phase-dependent coupling). A concise lemma or constructive argument would aid future extensions.
2. Type-1 vs. Type-2 mapping—intuition and limits. Provide a compact physical picture for why phase sensitivity redistributes among ports differently in the two mappings and clarify any fundamental tunability limits implied by the core equations.
3. Hardware validation of phase-setting formulas. Add calibration plots that (i) sweep θ_{MZI2} to recover the predicted $|A|^2$ mapping, and (ii) validate the arctan relation for $\phi_{MZI3} - \phi_{MZI2}$ versus $(|t|, |r|, \phi_{rt})$.
4. Origin and mitigation of MZI3 asymmetry. Quantify the observed imbalance (effective reflectivity error or internal phase bias), include a simple correction model, and comment on whether dual-port calibration would remove the bias.
5. Metrological figure of merit. Report both per-trial and per-detected-photon FI (accounting for routing to the ancilla) and discuss estimators under realistic detector inefficiency; briefly compare to unitary MZI NOON interferometry.
6. Programmable filtering without destruction. Consider adding a concept figure or brief demonstration of reinjecting the ancilla output back into the mesh to realize adaptive/conditional POVMs or multiplexed estimation.
7. Scalability of the CPA block. Provide resource scaling (MZIs, phase shifters, heater power) for embedding k non-unitary 2×2 blocks within an m -mode mesh. A small table or asymptotic comment would be helpful.
8. Robustness to spectral/temporal mismatch. Quantify how a small distinguishability ϵ degrades deterministic single-photon and probabilistic two-photon absorption signatures and FI (e.g., perturbative curves/fits).
9. Approach to the theoretical bound $|A|^2 \leq 0.5$. Show the closest experimental approach to 0.5, with an uncertainty budget identifying limiting factors (mesh errors, heater range, coupler mismatch).
10. Error model and Bhattacharyya overlap. Attribute the deviation from unity over-

lap to specific error sources—phase-setting errors, coupler imbalance, detector-efficiency mismatch, multi-pair contamination—via a simple linearized propagation or bootstrap.

Minor comments and clarifications

11. Notation consistency. Unify the use of φ_T (internal vs. relative phase) and remind readers of the $\varphi_T = 0$ gauge choice.
12. Figure captions (Type-2). In figures showing Type-2 behavior, add the explicit mapping from φ_T to the relative phase at the MZI3 inputs (pointer to the relevant equation/derivation) in the caption.
13. SPDC source details. Bring singles, heralding rates, pair rate, pump power, and accidental-coincidence subtraction into the main text or Methods (you already provide $g(2)$ metrics).
14. Detector model. State SNSPD system detection efficiency, timing jitter, and dark-rate figures; confirm whether efficiencies were used to correct all probability bars.
15. Mesh calibration/stability. Report phase-stability metrics (rms drift over 1 h) and whether PID bandwidth limited any scans.
16. Clements mesh output phases. Clarify whether the observed $\delta_2 - \delta_1 = \pm\pi$ relation is intrinsic to the 3-MZI CPA layout or a convention of your decomposition.
17. Ancilla labeling. In single-photon plots, label the ancilla axis as “absorbed (ancilla)” and explicitly remind readers that “absorption” corresponds to routing to the ancilla.
18. NOON periodicity. Add a sentence in the main text that π periodicity arises from the $|2,0\rangle \pm e^{i2\varphi}|0,2\rangle$ phase.
19. Imbalance inset. Where discussing redistribution between $|0,2,0\rangle$ and $|1,1,0\rangle$, include an inset curve showing the effect of a $\pm\epsilon$ MZI imbalance.
20. Reference hygiene. Where citing software (e.g., QOptCraft and the modified interferometer package), include version tags/commit hashes.

(Remarks on code availability)

Reviewer #3

(Remarks to the Author)

(Remarks on code availability)

Reviewer #4

(Remarks to the Author)

The manuscript by Krishna et al. investigates Coherent Perfect Absorption (CPA) in the quantum regime by implementing a reconfigurable photonic network capable of probing quantum interference effects in lossy systems. The authors design and experimentally realize an integrated beam splitter with tunable losses, transmission, and reflection coefficients on a silicon-on-insulator platform. The effective absorption is controlled via adjustable coupling to an auxiliary waveguide, enabling dynamic tuning of the system's dissipation properties. The chip's functionality is verified using both single- and two-photon input states. Importantly, the experiments demonstrate a clear advantage of two-photon interferometric measurements over single-photon approaches, thereby highlighting a potential metrological application of the proposed platform.

Assessment of Impact:

The study addresses a timely and conceptually relevant topic at the interface of quantum optics, integrated photonics, and non-Hermitian physics. The implementation of reconfigurable CPA on a silicon photonic chip represents a significant step toward control of coherent loss mechanisms in quantum systems. The experimental validation using quantum states of light adds substantial weight to the manuscript.

Support of claims:

The authors provide a thorough description of how the beam splitter parameters are implemented within the integrated circuit and convincingly demonstrate the tunability of these parameters in the experiments.

The main claim that “all previously reported quantum CPA effects, including deterministic absorption, anti-coalescence, photon bunching and probabilistic two-photon absorption” can be observed within this architecture is supported for deterministic absorption, photon bunching, and probabilistic two-photon absorption.

However, anti-coalescence of photons cannot be supported by the data shown in Fig. 5 (at $\varphi = 0.5\pi$ or 1.5π in Fig. 5b). Anti-coalescence, as first reported by Vest et al. [12], becomes more pronounced with increasing loss. In stark contrast, the authors identify the low-absorption case as evidence for anti-coalescence. Consequently, these data cannot distinguish genuine anti-coalescence from the trivial case of Hong–Ou–Mandel interference in a beam splitter with full transmission/reflection and are more indicative of the latter scenario. A suggestion for experimental evidence is provided in the Methodology section.

Another claim that remains unsupported concerns scalability. Since the title does not specify the type of quantum light, the

authors should clarify why measurements with single- and two-photon states are sufficient to represent the general framework. Additionally, it remains unclear how the resources required to implement tunable loss scale when extending the system to a 3×3 network (or larger). The authors should discuss whether the necessary number of auxiliary waveguides or the constraints of a planar geometry impose limits on scalability or reconfigurability.

Methodology and Reproducibility:

While the general experimental procedures and data presentation are clear and appear reproducible, several methodological aspects require clarification:

For the calculation of the Bhattacharyya coefficients, the summation indices are not explicitly defined. Presumably, they run over the set of basis states, but this should be stated unambiguously.

Typically, quantum interference effects such as photon bunching and anti-coalescence are characterized through Hong–Ou–Mandel–type experiments, contrasting the quantum interference of indistinguishable photons with a classical baseline of distinguishable ones. This distinction is particularly relevant to the manuscript’s claim of observing anti-coalescence, which currently lacks sufficient experimental evidence or discussion.

The consequences of constraining the full eight-parameter space of a general 2×2 transfer matrix to only three effective parameters $|t|$, ϕ_{rt} , and $|A|$ are not addressed. The authors should elaborate on how/if this restriction limits the accessible CPA conditions and what physical insights might be lost or preserved. What is the motivation of only discussing two types of beam splitters?

The experimental network comprises eight waveguides, whereas the theoretical model requires only three. The manuscript should clarify the rationale for this expanded implementation — in particular, why three additional beam splitters are placed around MZIs 1–3 and how these affect the system’s overall transfer characteristics or tunability. If these additional couplers are introduced for calibration, mode matching, or symmetry reasons, this should be explicitly discussed.

Aside from these points, the remaining methodological descriptions are well presented and should allow independent reproduction of the main experimental results.

Clarity of Report:

The manuscript would benefit from several improvements to visual clarity and presentation consistency:

Figure 1 should be expanded to illustrate the main concept of CPA—starting from the classical counterpropagating-wave picture, transitioning to propagating waves in integrated systems, and extending to the quantum regime (single- and two-photon cases). This conceptual sketch would help readers grasp the hierarchy of effects and claims.

The color scheme requires attention: the data in Figure 3 (top row) should use a consistent palette with Figure 4 to maintain visual coherence.

Consider adding a conceptual comparison graphic contrasting classical and quantum CPA.

In Figure 1c, the difference between green and red shades should be more pronounced for better readability.

The inclusion of the SPDC source in Figure 1c as part of the integrated setup contradicts Figure 2, where it is shown as an external free-space source; this inconsistency should be corrected or explained.

The text should include explicit references to each subpanel of Figures 4 and 5 at the points where they are discussed, to enhance readability and guide the reader through the results.

Recommendation:

The manuscript presents a solid experimental advance with clear relevance to integrated quantum photonics and should merit publication after mayor revision addressing the points raised.

(Remarks on code availability)

Version 1:

Reviewer comments:

Reviewer #2

(Remarks to the Author)

I have reviewed the authors’ rebuttal and revised manuscript/SI. The revision substantially strengthens the work in clarity, conceptual framing, and experimental validation, and it addresses my major concerns. In particular, the authors now clearly justify the ancilla requirements (including the general rank condition), clarify the physical distinction between Type-1 and Type-2 mappings and associated constraints, provide experimental validation of the closed-form programming relations, and give a plausible quantitative explanation for the remaining small discrepancies (e.g., the diagnosed MZI3 asymmetry). The manuscript is now in a publishable state.

(Remarks on code availability)

n.a.

Reviewer #3

(Remarks to the Author)

I co-reviewed this manuscript with one of the reviewers who provided the listed reports. This is part of the Nature

Communications initiative to facilitate training in peer review and to provide appropriate recognition for Early Career Researchers who co-review manuscripts.

(Remarks on code availability)

Reviewer #4

(Remarks to the Author)

(Remarks on code availability)

Response to Referees' Report

We appreciate the time and effort that the editors and reviewers have put into the review process for our manuscript entitled "Emulation of Coherent Absorption of Quantum Light in a Programmable Linear Photonic Circuit".

Color legends for the revised manuscript:

All revisions are highlighted in colour in the revised version:

Reviewer 1 (blue); Reviewer 2 (orange); Reviewer 4 (violet); Global or multi-reviewer changes (red); Post-submission correction note (cyan).

Each change is also tagged in the \LaTeX source using comments (e.g. `% R1C3`) corresponding to the reviewer and comment numbers.

Note on a post-submission correction to the data analysis

After re-examining our photon-number-resolving data following submission, we identified and corrected a normalization error that affected the reconstructed probabilities of two-photon Fock states in the NOON-state experiment. Specifically, in our detection scheme, each output mode of the lossy beam splitter circuit block is followed by an extra 50:50 Mach-Zehnder interferometer (MZI) for photon number resolution. For events in which both photons exit through the same output mode, corresponding to the Fock states $|200\rangle$, $|020\rangle$, and $|002\rangle$, the two photons are split randomly by this 50:50 MZI. Consequently, only half of these events (those in which the photons are separated between the two output arms of the MZIs) are registered as coincidences between the two corresponding detectors, giving an effective detection efficiency of 50% for same-mode two-photon events. So the experimentally measured coincidence counts corresponding to these Fock states have to be multiplied by a factor of 2 to extract the actual counts. In the originally submitted version, this factor of two was not applied, leading to an underestimation of these probabilities. In the corrected analysis, we consistently multiply the coincidence counts corresponding to these events by 2 and recompute all derived quantities (Fock-state probabilities, Bhattacharyya overlaps, and classical Fisher information), updating the corresponding figures and text.

This correction introduces minor quantitative adjustments while leaving all qualitative conclusions unchanged:

- The reconstructed two-photon output distributions in Figs. 5 and 6 now show slightly higher weights for $|200\rangle$, $|020\rangle$, and $|002\rangle$, yielding closer agreement between experiment and theory for both beam splitter types. The data points for the $|110\rangle$ and $|002\rangle$ curves in particular now align more accurately with the theoretical predictions.
- The Bhattacharyya coefficients show a significant improvement, reaching values above 0.93 for all configurations (previously above 0.89). Across all absorption and phase settings in

the revised manuscript, the average Bhattacharyya overlap is 0.9850 ± 0.0122 for the Type-1 configuration and 0.9910 ± 0.0092 for the Type-2 configuration.

- The Fisher information (FI) analysis has been refined to distinguish between the *per-Fock-state* and *summed* contributions. The FI values presented in the figures correspond to the *per-Fock-state* quantities,

$$F_{|\psi_i\rangle}^{\max} = \max_{\phi} \left[\frac{1}{P_{|\psi_i\rangle}(\phi)} \left(\frac{dP_{|\psi_i\rangle}(\phi)}{d\phi} \right)^2 \right],$$

while the total phase sensitivity of the measurement is characterized by the *summed* Fisher information,

$$F_{\text{tot}}(\phi) = \sum_i \frac{1}{P_{|\psi_i\rangle}(\phi)} \left(\frac{dP_{|\psi_i\rangle}(\phi)}{d\phi} \right)^2.$$

An additional equation has been inserted in the revised text to clarify this distinction. The maximum total Fisher information obtained from the NOON-state measurements is now $F_{\text{tot,max}} \approx 3.35$ for the Type-1 (asymmetric) configuration and $F_{\text{tot,max}} \approx 3.4$ for the Type-2 (symmetric) configuration, instead of the previously quoted per-Fock-state value of 3.7. These summed values, which are the relevant quantities for comparison with the fundamental limits, still exceed the shot-noise limit ($F_{\text{SQL}} = 2$) and remain close to the Heisenberg limit ($F_{\text{HL}} = 4$).

- The single-photon data and trends are unaffected. For consistency, we now report the *total* Fisher information for the single-photon case, $F_{\text{tot,max}} = 1$, rather than the per-mode value quoted earlier.

In summary, the corrected analysis yields improved quantitative agreement between experiment and theory while preserving all the central physical conclusions of the work. We apologize for this oversight and hope that this clarification assists in evaluating the revised manuscript.

Response to the Reviewers

Reviewer 1

Reviewer Comment 1.0 — In this manuscript, the authors report on an experimental effort to demonstrate nonunitary transformations to emulate coherent absorption of quantum light. Their approach is based on a programmable linear photonic platform where loss is implemented through ancilla modes.

The authors demonstrate the well-established concept of coherent perfect absorption (CPA) using a versatile programmable photonic circuit. Their platform can tune the all-physical parameters to implement PCA in a broad range of single-photon and two-photon quantum states. This is a very appealing work, and I am positive in recommending this paper for publication in Nature Communications, provided the comments detailed below are addressed.

Reply: We sincerely thank the reviewer for the positive and encouraging assessment of our work. We are delighted that the reviewer recognizes the significance of implementing nonunitary transformations on a programmable photonic platform and appreciates the versatility of our approach for exploring coherent perfect absorption (CPA) with quantum light.

In the revised manuscript, we have carefully addressed all the specific comments raised below and improved the clarity of presentation. We are grateful for the reviewer’s constructive feedback and for recommending our work for publication in *Nature Communications*.

Reviewer Comment 1.1 — Suggestions for improving the presentation of the paper.

In general, the authors present very convincing results unequivocally demonstrating the effects of PCA. However, I find the writing of the manuscript quite technical, making the concepts and the description of the device operations hard to follow. As it is written, the paper would perfectly fit for a more specialized Journal. For instance, the authors did a good job describing how coherent absorption arises in classical systems. Specifically they State:

“Classically, coherent absorption arises when counterpropagating waves form a standing wave, with the absorption determined by the absorber’s position within the resulting field profile. Complete absorption or coherent perfect absorption (CPA) occurs at antinodes, full transmission at nodes, and partial absorption elsewhere.”

This statement is crucial and clearly explains the phenomenon of classical coherent absorption. I suggest the authors to elaborate the concepts for the quantum cases in a similar fashion and perhaps accompanying such explanations with a couple of figures visually illustrating the physical processes.

Reply: We thank the reviewer for the constructive suggestion. We agree that presenting the quantum coherent absorption process in a more intuitive manner, closely analogous to the classical standing-wave picture, helps readers better appreciate the underlying physics.

While the original manuscript already contained a small conceptual description of this mechanism (which was intentionally limited mainly due to word limit concerns) in the *Results* section, in response to the reviewer’s comment we have further refined and expanded this discussion to improve clarity and accessibility. In the revised manuscript, we describe how the input quantum state is projected, via quantum interference at MZI_1 , onto two orthogonal superposition modes, how the subsequent coupling of the post- MZI_1 lower rail to the ancilla mode and the second interference at MZI_3 determine the output state probabilities, and how these processes together realize phase-controlled absorption/quantum state engineering. In this framework, the relative phase between the input modes governs the first interference process at MZI_1 , while the intrinsic beamsplitter phase ϕ_{rt} controls the second interference of the surviving light at MZI_3 . This provides a clear quantum analogue of the classical node–antinode picture and explains the coherent absorption behavior observed for both single-photon and two-photon (NOON-state) inputs.

In direct response to the reviewer’s suggestion, we have also introduced a new Fig. 1 in the revised manuscript (which we also present here as Fig. 1), which schematically illustrates this interference-based picture of quantum coherent absorption in a single, unified visual framework. The figure highlights the role of the two interference stages, the programmable coupling to the ancilla mode, and their combined effect on phase-controlled absorption, thereby making the analogy to the classical standing-wave picture more transparent.

As additionally suggested by the reviewer, we have included a concise summary of this conceptual framework in the *Introduction* to guide the reader before the detailed experimental and theoretical

discussion. More formal theoretical descriptions of the quantum state evolution have been added to Supplementary Section S4, and for further theoretical context we refer the reader to Vetlugin *et al.* [1], which is cited at the relevant points in both the main text and the Supplementary Information.

Reviewer Comment 1.2 — In the present form, the manuscript just describes the experimental platform, the quantum states used and the results. That is, it lacks a physical description of the effects and how specifically the states evolve through the system.

Reply: We appreciate the reviewer’s valuable comments and have further improved the description of how the quantum states evolve through the programmable CPA circuit. As mentioned in our response to Comment 1.1, the relevant parts of the *Results* section have been refined to present this evolution more clearly for both the single- and two-photon experiments.

In the single-photon case, the revised text outlines how MZI_1 projects the input dual-rail state into symmetric and antisymmetric superposition modes (the upper rail after MZI_1 couples to the antisymmetric superposition of the input modes and the lower rail to the symmetric superposition), with the lower rail coupling to the ancilla at MZI_2 and the remaining amplitude in the upper and lower rails undergoing a second quantum interference at MZI_3 (dictated by ϕ_{rt}) governed by the internal phase ϕ_{rt} . This stepwise description clarifies how phase control determines absorption and transmission in the quantum regime.

For the NOON-state experiment, we have expanded the discussion to describe the two-photon transformation more transparently: after MZI_1 , the input state $|\Psi_{\text{NOON}}\rangle = (e^{i\phi}|20\rangle - |02\rangle)/\sqrt{2}$ is transformed onto a superposition of $|2_+\rangle = (|20\rangle + |02\rangle)/\sqrt{2}$ and $|11\rangle$ states (both these states defined w.r.t to the upper and lower rails of the MZI_1 output), with ϕ setting their relative amplitudes. The portion of light propagating in the lower arm couples to the ancilla at MZI_2 , and the surviving amplitudes interfere again at MZI_3 to yield the observed phase-dependent probabilities of single-photon loss, two-photon loss, bunching, and anti-coalescence. This extended explanation links the experimental observations directly to the underlying two-photon interference process.

For completeness, we refer the readers to the supplementary section S4 and to the discrete-variable coherent absorption framework developed by Vetlugin *et al.* [1], which provides the full theoretical treatment of these transformations and is cited at all relevant points in the manuscript. The revisions therefore, strengthen the physical narrative of the experiment while maintaining the theoretical discussion within the intended scope of this work.

Reviewer Comment 1.3 — Additionally, why the authors decided to use the Clements configuration?

Reply: Our decision to use the Clements configuration for the 8-mode universal interferometer is motivated by the following practical and conceptual reasons:

1. **Minimal optical depth and reduced propagation loss.** The Clements rectangular mesh achieves the minimal possible optical depth for an $N \times N$ universal interferometer (depth = N vs. $2N - 3$ for the Reck triangular mesh), while using the same minimal number of beam splitters [2, 3]. This shorter depth directly reduces propagation loss which is crucial for experiments with single photon and two photon states.
2. **Improved tolerance to unbalanced loss.** As shown by Clements *et al.* [2], the symmetric layout of the rectangular mesh leads to better matching of path lengths and makes the implemented

transformations significantly more robust to beam-splitter–induced, unbalanced loss than in the Reck architecture. This enhanced loss tolerance is crucial for faithfully realizing the target non-unitary CPA transformations in the presence of inevitable fabrication imperfections.

- 3. Compatibility with programmable photonic hardware and calibration.** Rectangular meshes based on the Clements architecture have become a standard choice for programmable linear photonic processors [4, 5] because of their regular layout and ease of control and calibration. Our 8-mode chip is designed along these lines, allowing us to directly leverage existing characterization and tuning procedures [6], which in turn simplifies the accurate programming of the CPA circuit.

Reviewer Comment 1.4 — Why the Fisher information is a good estimator to quantify the phase sensitivity of the device? The authors must explain in a few words the physical meaning of Eq. (6), not just state what it does?

Reply: We agree that its important to clarify the physical meaning of the Fisher information (FI) in our analysis.

In our measurements, the input phase ϕ is a controlled parameter that we vary experimentally, and the purpose of using FI is to quantify how sensitively the output photon-number distribution responds to changes in ϕ . The FI provides a standard way to capture this response: it is large when a small variation of ϕ produces a strong change in the detection probabilities $P_{|\psi_i\rangle}(\phi)$, and it vanishes when these probabilities are flat in ϕ . Equation (6) quantifies this phase sensitivity for a single output Fock state, while Eq. (7) gives the total sensitivity by summing over all accessible outcomes.

Although FI is often introduced in the context of estimating an unknown phase, it more generally characterises the intrinsic responsiveness of an interferometer to phase variations through the Cramér–Rao bound, which relates FI to the ultimate attainable precision for any photon-counting measurement scheme. For any unbiased estimator of ϕ based on repeated measurements with a fixed detection scheme (here, photon counting in the Fock basis), the phase variance is bounded as $\Delta\phi^2 \geq 1/[\nu F_{\text{tot}}(\phi)]$, where ν is the number of experimental repeats [7]. The peak value of $F_{\text{tot}}(\phi)$ therefore provides a natural figure of merit for assessing the phase sensitivity of our programmable lossy beam splitter.

In response to the reviewer’s suggestion, we have added a concise explanation of this physical interpretation in the revised manuscript immediately after Eqs. (6) and (7).

Reviewer Comment 1.5 — The results presented in the manuscript are compelling, but the overall clarity and structure need to be improved before the work can be considered for Nature Communications.

Reply: We thank the reviewer for the positive assessment of our results and for the constructive feedback regarding the presentation. In response, we have substantially improved the clarity and structure of the manuscript by refining several explanations and adding additional context wherever needed. We believe that the revised version is now clearer, more accessible, and better aligned with the high standards of *Nature Communications*.

Reviewer 2

Reviewer Comment 2.0 — This manuscript demonstrates a programmable, on-chip emulation of coherent absorption (CPA) by embedding a lossy 2×2 transformation within a larger unitary via a single ancilla mode and providing closed-form settings that map absorptivity and internal phase directly to MZI controls. The experiments with single- and two-photon probes reproduce the predicted phase dependences and output distributions with high overlap, and the reported Fisher-information enhancement (approaching the $N=2$ Heisenberg limit) substantiates the metrological claims. Beyond prior fixed CPA implementations, the work elevates CPA to a programmable primitive that is firmly grounded in quasi-unitary dilation and Clements-mesh synthesis, and should be impactful for programmable non-unitary photonics, POVM engineering, and multiplexed quantum sensing. The methodology is sound and meets community standards, the conclusions are supported by the data. I see no publication-blocking flaws, only minor clarifications that would strengthen transparency: a brief on-chip calibration validating the closed-form settings, a compact error budget for the residual distribution mismatches (including MZI imbalance and detector effects), and basic detector specifications alongside per-detected-photon FI. Reproducibility is good at the level of theory and procedures; to make it turnkey for others, I recommend sharing representative raw time-tag histograms with analysis scripts and a short, version-pinned code/README to regenerate key plots. I find the work technically solid, original in its programmable framing, and suitable for publication after minor revision.

Reply: We thank the reviewer for the clear summary of our work and for the positive assessment. We have carefully considered all comments and, guided by the constructive suggestions, we have undertaken an extended revision of both the main manuscript and the Supplementary Information.

In addition, to further strengthen transparency and reproducibility, we will submit a complete data-and-code package together with the revised submission. This package will contain (i) all raw experimental data files exported as `.csv` tables, (ii) the full set of plotting/analysis scripts used to generate the manuscript figures, and (iii) a dedicated README / quick-start guide describing the repository structure, software dependencies, and clear instructions for reproducing each figure.

Regarding the raw coincidence data: in our measurements we do not record and store full time-tag histograms directly from the Time Tagger software. Instead, we use a custom acquisition script that controls the time tagger and directly accumulates coincidence counts within a predefined coincidence window (2 ns) as a function of the swept experimental parameter(s). We provide the full raw coincidence data as structured `.csv` files (one-to-one with the measurement sweeps) together with the complete analysis/plotting code to reconstruct all processed quantities and figures.

Reviewer Comment 2.1 — Conceptual scope of the non-unitary emulator. Please articulate conditions under which a single ancilla suffices for port-symmetric lossy beam splitters and when additional ancilla modes become necessary for more general non-unitary maps (e.g., asymmetric loss, phase-dependent coupling). A concise lemma or constructive argument would aid future extensions.

Reply: We thank the reviewer for this important conceptual question. In the revised manuscript, we now make explicit the precise conditions under which a single ancilla mode suffices for emulating a lossy 2×2 transformation, and when additional ancilla modes become necessary. Additionally, we provide a detailed analysis regarding the same in supplementary Sec. S1.2

Our programmable CPA block implements the class of *reciprocal, port-symmetric lossy beam splitters*, whose transfer matrix can be written as

$$T = \begin{pmatrix} t & r \\ r & t \end{pmatrix}.$$

Port symmetry implies identical coupling of both ports to a single effective loss channel. As shown in Supplementary Sec. S1, this leads to the constraints

$$|t|^2 + |r|^2 + |A|^2 = 1, \quad 2|t||r| \cos \phi_{rt} = tr^* + rt^* = |A|^2.$$

Together, these relations fully determine the loss contribution,

$$I - T^\dagger T = |A|^2 \begin{pmatrix} 1 & 1 \\ 1 & 1 \end{pmatrix},$$

which is a positive semidefinite matrix of rank 1. Since the rank of a positive semidefinite matrix equals the number of its nonzero singular values, this implies

$$\text{rank}(I - T^\dagger T) = 1,$$

i.e., exactly one singular value of T differs from unity.

Within the quasi-unitary dilation framework, the minimal number of ancilla modes required to embed a passive non-unitary transformation equals the number of singular values of T that are strictly smaller than 1, or equivalently $\text{rank}(I - T^\dagger T)$ [8]. Therefore, a single ancilla mode is both necessary and sufficient for the port-symmetric CPA transformations considered here, and the map can be embedded in a 3×3 unitary acting on $\{\text{signal}_1, \text{signal}_2, \text{ancilla}\}$.

For more general non-unitary transformations—such as those with asymmetric loss, multiple independent absorption channels, or phase-dependent coupling— $I - T^\dagger T$ is generically full rank. In such cases, both singular values of a 2×2 lossy matrix are non-unitary, and the minimal dilation dimension increases accordingly. In general, the required number of ancilla modes is

$$m_{\text{anc}} = \text{rank}(I - T^\dagger T),$$

so that two ancilla modes are required when both singular values differ from unity.

To clarify these points, we have added a brief explanation in the second last paragraph of the “Quasiunitary decomposition scheme” subsection in the Methods section of the main text, and expanded Supplementary Sec. S1.2 with an explicit derivation of the loss matrix $I - T^\dagger T$, its rank, and the corresponding minimal ancilla dimension, together with a short discussion of the general (asymmetric) lossy beam splitter case.

Reviewer Comment 2.2 — Type-1 vs. Type-2 mapping—intuition and limits. Provide a compact physical picture for why phase sensitivity redistributes among ports differently in the two mappings and clarify any fundamental tunability limits implied by the core equations.

Reply: We thank the reviewer for this insightful comment. While the original manuscript established the formal mapping between the lossy beam-splitter parameters and the interferometer settings, it did not include a detailed analytical discussion of the redistribution mechanism. In response to the reviewer’s

comment, we have now added a new Supplementary Sec. S4.4 providing an explicit, equation-based treatment of this mechanism, and have revised the main text accordingly.

The redistribution of phase sensitivity among the output ports is governed by the relative phase between the two fields interfering at MZI₃. From the analytic mapping between the target lossy beam splitter parameters $(t, r, \phi_{rt}, |A|^2)$ and the MZI settings, main text Eq. (3) gives

$$\phi_{\text{MZI3}} - \phi_{\text{MZI2}} = \arg\left(\frac{t+r}{t-r}\right) + \frac{\theta_{\text{MZI2}}}{2} + \frac{\pi}{2} = \text{sgn}(\cos \phi_{rt}) \tan^{-1}\left(\frac{2|t||r| \sin(\phi_{rt})}{|t|^2 - |r|^2}\right) + \frac{\theta_{\text{MZI2}}}{2} + \frac{\pi}{2}, \quad (1)$$

where the sign function is defined as

$$\text{sgn}(x) = \begin{cases} +1, & x > 0, \\ -1, & x < 0, \end{cases} \quad (2)$$

with $\theta_{\text{MZI2}} = 2 \cos^{-1}(\sqrt{2|A|^2})$ (Eq. (2) in the main text), and the port-symmetric lossy-beamsplitter constraints

$$2|t||r| \cos \phi_{rt} = \pm |A|^2, \quad |A|^2 = 1 - |t|^2 - |r|^2, \quad (3)$$

(Eqs. (9),(10) in the main text).

It is convenient to isolate the interferometric phase at the inputs of MZI₃ by rearranging Eq. (1):

$$\Delta\phi_{\text{in}} = \phi_{\text{mode1}} - \phi_{\text{mode2}} = \phi_{\text{MZI3}} - \left(\phi_{\text{MZI2}} + \frac{\theta_{\text{MZI2}}}{2} + \frac{\pi}{2}\right) = \text{sgn}(\cos \phi_{rt}) \tan^{-1}\left(\frac{2|t||r| \sin(\phi_{rt})}{|t|^2 - |r|^2}\right), \quad (4)$$

where $\Delta\phi_{\text{in}}$ is the physical relative phase between the two optical fields immediately before the final interference at MZI₃. As shown explicitly in Supplementary Sec. S4.4, with MZI₃ fixed at a balanced (50:50) configuration, this phase enters the interference through a $\sin \Delta\phi_{\text{in}}$ dependence, and therefore directly governs how the surviving (non-ancilla) amplitude and phase sensitivity are distributed between the two signal outputs.

Type 1 (fixed $\phi_{rt} = \pi$): Here $\sin \phi_{rt} = 0$, so the arctangent term in Eq. (4) vanishes and

$$\Delta\phi_{\text{in}} = 0 \pmod{\pi}, \quad (5)$$

independent of $|A|^2$. Thus, tuning $|A|^2$ does not change the relative phase governing the MZI₃ interference, but only rescales the amplitudes reaching it (via MZI₂). This leads to a uniform reduction of interference visibility and hence a simultaneous suppression of phase sensitivity across all output modes as $|A|^2$ is reduced, consistent with the behaviour reported in the main text and figures.

Type 2 (symmetric $|t| = |r|$): Here $|t|^2 - |r|^2 = 0$ in Eq. (4), so

$$\Delta\phi_{\text{in}} = \text{sgn}(\cos \phi_{rt}) \tan^{-1}\left(\frac{2|t||r| \sin(\phi_{rt})}{|t|^2 - |r|^2}\right) = \pm \frac{\pi}{2} \pmod{\pi}, \quad (6)$$

with the sign set by $\sin \phi_{rt}$. Importantly, in the Type 2 mapping the internal phase ϕ_{rt} must vary with $|A|^2$ due to the constraints in Eq. (3). Consequently, although the interference at MZI₃ occurs at fixed quadrature, tuning $|A|^2$ modifies how the phase-dependent amplitudes interfere, thereby redistributing the surviving amplitude and phase sensitivity from the ancilla to the signal ports, and, in the multiphoton case, among different output Fock states with signal-mode components.

Fundamental tunability limits: The accessible parameter range of the effective lossy beam splitter is constrained by the physicality conditions in Eq. (3), which enforce both passivity and port symmetry.

In particular, the relations $|A|^2 = 1 - |t|^2 - |r|^2$ and $2|t||r| \cos \phi_{rt} = \pm|A|^2$ imply the bound $|A|^2 \leq 0.5$ (Eq. (11) in the main text), independent of the specific mapping. Within this bound, Type 1 and Type 2 mappings correspond to distinct paths through the same physically allowed parameter space: in Type 1, ϕ_{rt} is fixed while $|t|$ and $|r|$ vary with $|A|^2$, whereas in Type 2 the symmetry condition $|t| = |r|$ fixes the magnitudes and requires ϕ_{rt} to vary with $|A|^2$. These constraints fully determine the range over which the effective non-unitary beam splitter can be programmed and delimit the tunability of (t, r, ϕ_{rt}) achievable within a port-symmetric, passive implementation.

Reviewer Comment 2.3 — Hardware validation of phase-setting formulas. Add calibration plots that (i) sweep θ_{MZI_2} to recover the predicted $|A|^2$ mapping, and (ii) validate the arctan relation for $\phi_{\text{MZI}_3} - \phi_{\text{MZI}_2}$ versus $(|t|, |r|, \phi_{rt})$.

Reply:

Eqs. (2) and (3) in the main text provide closed-form analytical settings for the programmable MZIs used to implement the port-symmetric lossy beamsplitter.

Explicitly, Eq. (2) reads

$$\theta_{\text{MZI}_2} = 2 \cos^{-1} \left(\sqrt{2|A|^2} \right), \quad (7)$$

while Eq. (3) is given by

$$\phi_{\text{MZI}_3} - \phi_{\text{MZI}_2} = \frac{\theta_{\text{MZI}_2}}{2} + \frac{\pi}{2} + \text{sgn}(\cos \phi_{rt}) \tan^{-1} \left(\frac{2|t||r| \sin(\phi_{rt})}{|t|^2 - |r|^2} \right), \quad (8)$$

where $\text{sgn}(x) = +1$ for $x > 0$ and $\text{sgn}(x) = -1$ for $x < 0$.

In the experiments, the MZI phase settings were obtained by constructing a quasi-unitary extension of the non-unitary transformation and extracting the corresponding phases via a Clements decomposition. Therefore, the most direct validation of Eqs. (2) and (3) is to verify that the phase-shifter values predicted by these analytical expressions are identical to those obtained from the quasi-unitary embedding and Clements decomposition used to program the chip.

We emphasize that the experimental results reported in the manuscript themselves constitute a validation of Eqs. (2) and (3), since all measurements were performed using phase settings obtained from the Clements decomposition of the quasi-unitary extension, and the phase values predicted by the analytical formulas reproduce these Clements-derived settings with 100% accuracy.

We have explicitly performed this verification numerically over a large ensemble of physically allowed lossy beamsplitter parameters. For both Eq. (2) and Eq. (3), we obtain 100% agreement between the analytical predictions and the values extracted from the quasi-unitary+Clements procedure, with deviations below numerical precision. Representative correlation plots are shown in Fig. 2(a-b) in this review response document. We also provide the codes that we used to do this verification in the submitted compiled data/codes folder.

In addition, we provide an experimental validation of Eq. (2): with the input phase set to the condition of perfect absorption, the measured maximum ancilla-mode probability follows the predicted relation $P_{\text{ancilla}} = 2|A|^2$, yielding the same θ_{MZI_2} as given by Eq. (2). This ancilla-probability validation plot has been added to the Supplementary Information Sec S1.2 presented as Fig.S1.

Finally, while revisiting the derivation associated with Eq. (3) in response to this comment, we identified a subtle branch ambiguity in the trigonometric form of the phase relation. The final expression now includes an explicit $\text{sgn}(\cos \phi_{rt})$ factor, ensuring exact agreement with the quasi-unitary+Clements

decomposition across both physical branches of the lossy beamsplitter constraint. This correction improves the mathematical completeness of the derivation and does not affect any experimental results or conclusions.

Reviewer Comment 2.4 — Origin and mitigation of MZI₃ asymmetry. Quantify the observed imbalance (effective reflectivity error or internal phase bias), include a simple correction model, and comment on whether dual-port calibration would remove the bias.

Reply: We agree that a clearer discussion of the origin, quantification, and possible mitigation of the imbalance associated with MZI₃ is necessary.

In the revised Supplementary Information, we present a detailed analysis of the two-photon output statistics that motivates an effective two-port description of MZI₃ allowing for a small residual asymmetry (Supplementary Section S6). Within this framework, MZI₃ is described by a general 2×2 scattering matrix with slightly unequal amplitude products, an effect that becomes apparent for symmetric effective two-photon inputs to MZI₃ (specifically $|11\rangle$ and the symmetric NOON⁺ component), while leaving the antisymmetric NOON response largely unaffected.

This residual asymmetry is quantified by a single dimensionless parameter λ , extracted directly from the experimentally observed $|200\rangle$ – $|020\rangle$ imbalance, yielding $\lambda \simeq 1.1$ across both Type-1 and Type-2 configurations (Supplementary Secs. S6.4–S6.6). Using this parameter, we recompute the full two-photon output distributions. The resulting modified theoretical curves show good agreement with the experimental data, as illustrated in the new Supplementary Figs. S9 and S10 (also reproduced in this response (Figs. 3 and 4)), thereby supporting this interpretation. The codes used to do these calculations and plotting are presented in the data/codes folder that is submitted.

As discussed in Supplementary Section S6.7, plausible physical origins of such a residual asymmetry include fabrication-induced imbalance of the MMI couplers forming MZI₃, differential propagation loss between the two interferometer arms, and weak parasitic reflections or mode mismatch at waveguide junctions. Any of these effects can break the ideal port symmetry of the interferometer while remaining largely invisible to first-order transmission measurements.

An important practical aspect is that MZI₃ in the present circuit contains only a single internal phase shifter and is operated under the assumption of port symmetry. As a result, the available classical calibration degrees of freedom are insufficient to independently compensate for all possible internal asymmetries that affect two-photon interference [6]. While more elaborate calibration schemes could, in principle, provide additional information, they cannot fully eliminate asymmetries that originate from fixed fabrication imperfections in this architecture.

Potential routes to mitigating such effects therefore include improved fabrication accuracy of the MMI couplers and waveguides, as well as an extended MZI design providing additional phase-shifting degrees of freedom, together with a calibration protocol that constrains the full complex scattering matrix. An alternative approach is the use of two-photon interference itself as a characterization tool, since symmetric and antisymmetric quantum inputs directly probe the symmetry-sensitive combinations of scattering coefficients. However, such quantum-light-based calibration comes with practical drawbacks, including increased experimental overhead, reduced signal rates, and sensitivity to photon indistinguishability and loss, making it less suitable as a routine calibration method.

All these aspects—including the quantitative model, architectural constraints, and mitigation strategies—are discussed in detail in Supplementary Section S6.

Reviewer Comment 2.5 — Metrological figure of merit. Report both per-trial and per-detected-photon FI (accounting for routing to the ancilla) and discuss estimators under realistic detector inefficiency; briefly compare to unitary MZI NOON interferometry.

Reply: The Fisher information values reported in the main text correspond to the *maximum Fisher information* attainable for each programmed lossy beamsplitter matrix. For a given lossy beamsplitter, this maximum value is obtained by fitting the experimentally measured outcome probabilities acquired from a full input-phase sweep and evaluating the Fisher information as a function of phase. Consequently, there is a single maximum Fisher information value associated with each lossy beamsplitter matrix, which is the quantity reported in the main text.

For completeness, we note that for a fixed lossy beamsplitter matrix the total Fisher information generally varies with the input phase. In response to the reviewer’s comment, we now explicitly include in Supplementary Section S5 the Fisher information evaluated at all input phase values across the full phase sweep, illustrating how the maximum reported value arises from the experimental data.

Regarding realistic detector inefficiencies, all experimental probability distributions used to evaluate the Fisher information are normalized to account for unequal detector efficiencies, chip-fiber coupling differences and other loss channels. This normalization is performed using calibration data acquired prior to the experiment, in which the input light is routed sequentially and exclusively to each output channel and the corresponding counts are recorded over a fixed integration time. All data reported throughout the manuscript is therefore extracted from efficiency-corrected experimental probabilities.

Regarding the notion of a “per detected photon” Fisher information, we note that Fisher information is fundamentally defined for probability distributions obtained from measurements performed over a range of input-state phase values, rather than for individual detection events or photons. It is therefore not technically meaningful to assign a Fisher information value to a single detected photon. Instead, one can associate each detection event with the Fisher information of the interferometer configuration and input phase at which it occurred, as determined from the phase-dependent Fisher information already extracted for that interferometer from the experimental data.

For comparison with unitary MZI NOON interferometry, the $|A|^2 = 0$ setting in our device corresponds to the lossless (unitary) limit and thus provides the appropriate NOON-interferometric benchmark. In conventional lossless NOON interferometry, the highest phase sensitivity is obtained only for specific beamsplitter settings (e.g. near a balanced/symmetric configuration). In contrast, in our loss-emulating beamsplitter interferometer the coherent routing to, and measurement of, the ancilla mode enables high phase sensitivity to be accessed and redistributed across a broad range of lossy beamsplitter configurations, rather than being confined to a narrow set of lossless beamsplitter parameters.

Reviewer Comment 2.6 — Programmable filtering without destruction. Consider adding a concept figure or brief demonstration of reinjecting the ancilla output back into the mesh to realize adaptive/conditional POVMs or multiplexed estimation.

Reply: We thank the reviewer for this interesting and forward-looking suggestion. The possibility of routing and re-injecting ancilla modes to realize cascaded or conditional operations is indeed conceptually appealing within the general framework of ancilla-assisted linear optics.

The central aim of the present work, however, is to demonstrate the implementation and experimentally verified action of a single programmable non-unitary beam-splitter unit on a range of quantum input states. A detailed discussion or demonstration of cascaded architectures, conditional operation, or adaptive filtering would extend beyond this primary objective and shift the focus away from establishing the fundamental building block itself.

Nevertheless, we agree that such extensions represent an important direction. To reflect this, we have now explicitly highlighted in the Discussion section that the demonstrated programmable loss-emulating unit provides a versatile platform for simulating dissipative quantum channels and for realizing generalized measurements in an enlarged Hilbert space, which naturally connects to more advanced cascaded and conditional schemes.

Given the strict word and figure limits of the journal, and the already extensive experimental data required to substantiate the core results, we believe that a comprehensive treatment of these extensions lies beyond the scope of the present manuscript and is better suited to a dedicated future study. We have therefore retained a focused presentation centered on the single programmable unit and its experimentally verified action.

Reviewer Comment 2.7 — Scalability of the CPA block. Provide resource scaling (MZIs, phase shifters, heater power) for embedding k non-unitary 2×2 blocks within an m -mode mesh. A small table or asymptotic comment would be helpful.

Reply: We agree that it is important to discuss the resource scaling in detail. The required resources depend on how the k lossy 2×2 beam-splitter units are arranged within the interferometric network.

If the k units are arranged *in series*, implementing a sequential composition on the same two signal modes, the overall transformation remains 2×2 and the ancilla requirement does not scale with k : one ancilla suffices for port-symmetric (CPA-type) loss, while two ancilla modes are required for general asymmetric loss. The interferometric and electrical resources in this case scale linearly with k .

For a *multiport* arrangement, such as a Clements-type mesh implementing an m -mode transformation, the network contains

$$k = \frac{m(m-1)}{2}$$

tunable 2×2 couplers. Embedding the resulting non-unitary transformation T_{net} into a larger unitary requires

$$m_{\text{anc}} = \text{rank}\left(I - T_{\text{net}}^\dagger T_{\text{net}}\right)$$

ancilla modes (see Reply 2.1 and Supplementary Sec. S1.2), so the total unitary acts on $m + m_{\text{anc}}$ modes. A universal Clements mesh on these modes requires

$$N_{\text{MZI}} = \frac{(m + m_{\text{anc}})(m + m_{\text{anc}} - 1)}{2}$$

Mach–Zehnder interferometers. In our implementation, each MZI uses two thermo-optic phase shifters, giving $N_{\text{PS}} \approx 2N_{\text{MZI}}$ phase shifters.

With a typical electrical power P_π required for a π phase shift, a conservative upper bound on the total heater power scales as

$$P_{\text{tot}} \sim N_{\text{PS}} \times P_\pi,$$

corresponding to the worst case in which all phase shifters are driven near π . In practice, only a subset of heaters is tuned significantly for a given programmed transformation, so this estimate represents an upper bound rather than typical operating conditions.

These relations show that the interferometric and electrical resources scale polynomially with the number of modes, while the additional overhead specific to programmable loss is set by the rank of the net loss matrix rather than by the number of lossy beam-splitter units.

To make these scalings explicit in the manuscript, we added a brief asymptotic resource-scaling remark at the last paragraph of the “Quasiunitary decomposition scheme” subsection in the Methods section of the main manuscript and also the last paragraph of the supplementary section S1.2.

Reviewer Comment 2.8 — Robustness to spectral/temporal mismatch. Quantify how a small distinguishability ϵ degrades deterministic single-photon and probabilistic two-photon absorption signatures and FI (e.g., perturbative curves/fits).

Reply:

This effect is analysed in detail now in Supplementary Section S7. Increasing photon distinguishability suppresses phase-dependent two-photon interference, continuously converting coherent, interference-driven absorption into incoherent, phase-independent absorption. The impact of partial distinguishability can be quantified rigorously within our theoretical framework.

We model partial distinguishability by introducing a parameter $d \in [0, 1]$, where $d = 0$ corresponds to perfectly indistinguishable photons and $d = 1$ to fully distinguishable photons. For any two-photon output Fock state $|k\rangle$, the detection probability is written as

$$P_k(\phi; d) = (1 - d) P_k^{\text{ind}}(\phi) + d P_k^{\text{dist}}, \quad (9)$$

where $P_k^{\text{ind}}(\phi)$ contains all interference contributions, while P_k^{dist} is phase independent. This convex-mixture description of partial distinguishability follows the standard treatment of multi-photon interference [9].

In our experiment, the two-photon NOON state is generated via Hong–Ou–Mandel interference at the state-preparation stage. Increasing distinguishability reduces the fraction of events contributing to the NOON component and increases the weight of non-interfering contributions. Equation above captures this effect entirely at the level of the input state statistics, without modifying the device scattering matrix or its quasiunitary embedding.

All two-photon probabilities are computed using the same protocol as in the methods section of the main manuscript. For indistinguishable photons, $P_k^{\text{ind}}(\phi)$ is obtained from the full two-photon transformation. For fully distinguishable photons, the two photons are propagated independently using the corresponding single-photon transformation, and the resulting probabilities P_k^{dist} are combined incoherently.

Using experimentally relevant parameters, we evaluate all output probabilities as functions of the NOON phase ϕ , absorption strength $|A|^2$, and distinguishability d for both Type 1 and Type 2 beam-splitter implementations, considering $|A|^2 = 0$ and $|A|^2 = 0.5$. The results, shown in Supplementary Figs. S11 and S12 (also presented in this document as Figs. 5 and 6), demonstrate a continuous suppression of interference fringes with increasing d , while incoherent background probabilities remain finite. In the fully distinguishable limit, the phase dependence vanishes entirely.

From these probabilities we compute the total Fisher information, which decreases monotonically with increasing distinguishability and vanishes for $d = 1$, reflecting the loss of phase sensitivity. Since $\partial_\phi P_k$ arises solely from the interference term, it scales linearly with $(1 - d)$, leading to an approximately quadratic reduction of the Fisher information (Supplementary Fig S13 and presented in this document as Fig. 7). Importantly, absorption itself remains finite in the distinguishable limit, clearly separating incoherent absorption from phase-sensitive coherent absorption.

The complete theoretical and numerical analysis of photon distinguishability effects is provided in Supplementary Section S7.

Reviewer Comment 2.9 — Approach to the theoretical bound $|A|^2 \leq 0.5$. Show the closest experimental approach to 0.5, with an uncertainty budget identifying limiting factors (mesh errors, heater range, coupler mismatch).

Reply:

The point $|A|^2 = 0.5$ corresponds to the maximal intrinsic absorption setting of our effective lossy beam splitter (the coherent-absorption operating point), and in our platform there are *no additional limiting factors* that selectively prevent reaching the corresponding theoretical bounds at this value compared to other programmed absorption settings. Experimentally, we already report high-quality datasets at $|A|^2 = 0.5$ for *both* beam-splitter implementations (Type 1 and Type 2) and for *both* quantum input states (single-photon and two-photon NOON), with uncertainties that remain small and comparable to the rest of the parameter sweep. In all cases, error bars are derived from Poisson counting statistics with proper propagation through the normalization procedure (single-photon: Fig. 3 (main manuscript); two-photon: Figs. 4–5 (main manuscript)), and do not show any anomalous growth at $|A|^2 = 0.5$ relative to other absorption settings.

At the single-photon level, the maximal-absorption point exhibits the expected near-perfect absorption feature: at $\phi = \pi$ we observe nearly perfect absorption ($\sim 100\%$) for $|A|^2 = 0.5$, in agreement with theory (Fig. 3(b,e) (main manuscript)). This is precisely the regime where the circuit acts as a programmable single-photon coherent-absorption / state-filtering element.

At the two-photon level, the maximal-absorption point is also where the most distinctive interference-mediated absorption *regimes* occur. As discussed in the Results, at maximal $|A|^2$ and phases $\phi = 0, \pi, 2\pi$ the device exhibits *deterministic single-photon absorption* (one photon routed to the ancilla with near-unity probability), while at $\phi = \pi/2$ and $3\pi/2$ the system switches to *probabilistic two-photon absorption* with 50% probability and with the absorbed-photon bunching probability approaching 100% (Fig. 4(a–d) (main manuscript) for Type 1, and the corresponding behaviour in Fig. 5(a–d) (main manuscript) for Type 2). These “special regimes” are therefore not only accessible at $|A|^2 = 0.5$ in our system, but are in fact *most clearly manifested* there.

Finally, the overall agreement between experiment and theory remains high across the full $(\phi, |A|^2)$ space, including $|A|^2 = 0.5$: the Bhattacharyya overlaps exceed 0.93 across all absorption and phase settings (Figs. 4(e) and 5(e) (main manuscript)), confirming that the maximal-absorption point is implemented with comparably high fidelity. Consistently, the Fisher-information analysis shows pronounced features at high absorption (Figs. 4(f), 5(f) (main manuscript)), reflecting the enhanced phase sensitivity that accompanies these maximal-absorption interference regimes.

Reviewer Comment 2.10 — Error model and Bhattacharyya overlap. Attribute the deviation from unity overlap to specific error sources—phase-setting errors, coupler imbalance, detector-efficiency mismatch, multi-pair contamination—via a simple linearized propagation or bootstrap

Reply: We thank the reviewer for requesting further clarification regarding the origin of the deviation of the Bhattacharyya overlap from unity.

We find that the dominant contribution arises from the deviation of the experimentally measured $|200\rangle$ and $|020\rangle$ Fock-state probability curves from their ideal theoretical predictions. As discussed in detail in our response to Comment 2.4 and in Supplementary Section S6, these deviations are consistent with a small residual *port-asymmetry* of the effective 2×2 transfer matrix of MZI3. Plausible physical origins include fabrication-induced imbalance of the MMI couplers, differential propagation loss between the interferometer arms, and weak parasitic reflections or mode mismatch at junctions. Importantly,

such asymmetries can remain largely invisible under our standard single-port, intensity-only classical calibration [6], which fixes only a subset of the effective scattering parameters, yet they directly modify the coherent coefficient combinations governing two-photon interference (e.g. $t^2 + r^2$ versus $t'^2 + r'^2$), thereby producing unequal $|200\rangle$ and $|020\rangle$ probabilities. Since the Bhattacharyya overlap is sensitive to discrepancies across the full output probability distribution, this residual two-photon imbalance directly limits the achievable overlap even when all other Fock-state probabilities closely follow theory.

Additional, smaller deviations may arise from experimental imperfections related to polarization control. The multimode interference (MMI) couplers on the chip are designed to operate as ideal 50:50 beam splitters for transverse-electric (TE) polarized light, making accurate preparation of the input polarization crucial. On the detection side, the superconducting nanowire single-photon detectors (SNSPDs) are polarization sensitive, rendering the output polarization equally important. To mitigate this, we adjust the output fiber-based polarization controllers to maximize the detection efficiency in all channels and subsequently route the input light to each output channel sequentially; this dataset is then used for normalization.

Despite these precautions, the experiment relies on single-mode fibers to route light from the chip to the detectors. The polarization state in such fibers is sensitive to small mechanical perturbations, and accidental fiber movement in a shared laboratory environment, or vibrations from nearby instruments such as vacuum pumps, can introduce small, untracked polarization drifts between calibration and data acquisition. Additional minor deviations may also arise from very small temperature fluctuations of the SPDC crystal or the chip itself. Although both are highly temperature stabilized, localized heating can occur on the chip when nearby phase shifters are biased at high currents, potentially inducing transient unwanted phase shifts.

All these factors cannot be easily isolated from one another, but collectively affect the final photon counts. We therefore adopted an error analysis based on Poisson statistics, which is commonly accepted in the quantum optics community, and these errors are represented as error bars in all probability curve plots presented in the manuscript. Overall, the revised analysis demonstrates that the Bhattacharyya overlaps, exceeding 0.93 for all configurations, are primarily limited by the residual two-photon imbalance described in Supplementary Section S6, with all other effects contributing only minor corrections. We have now stated this attribution explicitly in the main text, in the *State fidelity and phase sensitivity* subsection of the NOON-state experiment section.

Reviewer Comment 2.11 — Notation consistency. Unify the use of f_{rt} (internal vs. relative phase) and remind readers of the $f_t = 0$ gauge choice.

Reply: We thank the reviewer for the careful reading. We have carefully checked the manuscript and confirm that the internal beam-splitter phase ϕ_{rt} and the input-state phase ϕ are used consistently throughout. The gauge choice $\phi_t = 0$, and hence $\phi_{rt} = \phi_r - \phi_t = \phi_r$, is explicitly stated at the beginning of the Results section where the lossy beam splitter is introduced.

To further improve clarity for the reader, we have added a brief reminder sentence in the main text explicitly distinguishing the fixed internal phase ϕ_{rt} from the externally scanned input-state phase ϕ .

Reviewer Comment 2.12 — Figure captions (Type-2). In figures showing Type-2 behavior, add the explicit mapping from φ_{rt} to the relative phase at the MZI3 inputs (pointer to the relevant equation/derivation) in the caption.

Reply: We thank the Reviewer for this suggestion. We have now explicitly added the mapping between the intrinsic beam-splitter phase ϕ_{rt} and the relative phase at the inputs of MZI₃, $\Delta\phi_{in}$, directly to the captions of the relevant figures in the main text.

Specifically, this clarification has been added to Fig. 4 (Fig. 3 in the submitted version) for the single-photon experiments, as well as to Fig. 5 and Fig. 6 (Figs. 4 and 5 in the submitted version) for the NOON-state experiments. In each case, the captions now explicitly state the corresponding fixed value of $\Delta\phi_{in}$ for Type-1 and Type-2 beam splitters, and refer the reader to Eq. 3 in the main text and to Supplementary Section S4.4 for the detailed derivation.

Reviewer Comment 2.13 — SPDC source details. Bring singles, heralding rates, pair rate, pump power, and accidental-coincidence subtraction into the main text or Methods (you already provide $g(2)$ metrics).

Reply: We thank the reviewer for the helpful suggestion. We have now added the following details of the SPDC source. The pump power is 70 mW, provided by a continuous-wave laser centered at 785 nm (driven at 220 mA). The single-path brightness is approximately 150 kHz, and the heralding efficiency is about 20 %, corresponding to a photon-pair generation rate of 30 kHz. The background noise and detector dark counts are sufficiently low that accidental coincidences are negligible when using a 2 ns coincidence window.

We have now also moved the characterization of the SPDC source from the Supplementary Information to the Methods section of the main manuscript. This includes the heralded $g^{(2)}$ measurement used to verify single-photon operation, the coincidence window definition employed throughout the experiments, and the spectral overlap characterization of the two SPDC outputs. The corresponding figures have been relocated accordingly (Now Figs. 7 and 8 in the main manuscript).

Reviewer Comment 2.14 — Detector model. State SNSPD system detection efficiency, timing jitter, and dark-rate figures; confirm whether efficiencies were used to correct all probability bars.

Reply: We thank the reviewer for the helpful suggestion. All superconducting nanowire single-photon detectors (SNSPDs) used in our experiment were provided by Single Quantum B.V. (a spin-off company co-founded by Prof. Val Zwiller). The detectors were fabricated in the same batch, resulting in highly uniform performance across all channels. The system detection efficiency of the SNSPDs exceeds 70%, with a timing jitter below 20 ps and a dark count rate of less than 10 cps.

We have added these details in the Method section of the main manuscript.

Reviewer Comment 2.15 — Mesh calibration/stability. Report phase-stability metrics (rms drift over 1 h) and whether PID bandwidth limited any scans.

Reply: We thank the reviewer for the helpful suggestion. In our experiment, the photonic chip is thermally stabilized using a TEC controller (TED 200C from Thorlabs). The controller stabilizes the on-chip heater resistance at the level of 1Ω , corresponding to a temperature stability of approximately $0.002 \text{ }^\circ\text{C}$. Once the chip is calibrated, this level of thermal control ensures excellent long-term stability. As further evidence, we refer the reviewer to our recent work (<https://arxiv.org/abs/2512.20273>), where a different experiment was performed on the same platform after a three-month interval, demonstrating consistent and stable performance.

We have added the description of the TEC controller in the supplementary section S3.4.

Reviewer Comment 2.16 — Clements mesh output phases. Clarify whether the observed $\delta_1 - \delta_2 = \pm\pi$ relation is intrinsic to the 3-MZI CPA layout or a convention of your decomposition.

Reply: The observed $\pm\pi/2$ relation between the output phases is not a general feature of the Clements decomposition of arbitrary unitary matrices. Rather, it arises specifically for the class of port-symmetric lossy beam-splitter matrices considered in this work, which, after quasi-unitary embedding and Clements decomposition, reduce to the three-MZI CPA layout shown in the manuscript. In this case, the phase relation reflects the physical constraints imposed by port symmetry and the embedding of loss via a single ancilla mode, rather than a decomposition convention.

Reviewer Comment 2.17 — Ancilla labeling. In single-photon plots, label the ancilla axis as “absorbed (ancilla)” and explicitly remind readers that “absorption” corresponds to routing to the ancilla.

Reply: We thank the Reviewer for raising this point. We would like to clarify that none of the single-photon plots use “ancilla” as an axis label. In all single-photon measurements, the horizontal axis corresponds to the input state phase ϕ (in units of π), while the vertical axis represents normalized detection counts. The ancilla mode appears only as a distinct output channel, representing the absorbed optical power, in panels where signal and ancilla mode outputs are compared, and not as an axis variable.

To avoid any possible ambiguity, we have clarified wherever appropriate that the ancilla mode represents the absorbed light, without excessive repetition.

Reviewer Comment 2.18 — NOON periodicity. Add a sentence in the main text stating that the π periodicity arises from the $e^{i2\phi}$ phase dependence of the $|20\rangle \pm e^{i2\phi}|02\rangle$ state.

Reply: We thank the Reviewer for this suggestion. We would like to note that the physical origin of the π -periodicity in the NOON-state interference is already stated explicitly in the final sentence of the third paragraph of the *NOON-state experiment* section (immediately preceding the *Type 1: Fixed $\phi_{rt} = \pi$* subsection). There, we clarify that, in contrast to the 2π -periodic fringes observed for single-photon inputs, the π -periodic oscillations directly reflect the two-photon path entanglement intrinsic to the NOON state (Eq. 5), i.e. the underlying $e^{i2\phi}$ phase dependence.

For clarity and to more directly address the Reviewer’s comment, we have made a slight wording refinement to this sentence to explicitly emphasize the $e^{i2\phi}$ phase dependence.

Reviewer Comment 2.19 — Imbalance inset. Where discussing redistribution between $|0,2,0\rangle$ and $|1,1,0\rangle$, include an inset curve showing the effect of a $\pm\varepsilon$ MZI imbalance.

Reply: In the originally submitted manuscript, a redistribution involving the $|020\rangle$ and $|110\rangle$ outcomes was attributed to a possible imbalance of MZI_3 . Following the post-submission correction described at the beginning of this response document, this interpretation has been revised based on a more detailed analysis of the experimental data, after which the $|110\rangle$ probability is found to be in very good agreement with the theoretical prediction.

As explained in our response to Comment 2.4 and in Supplementary Section S6, the dominant residual discrepancy in the revised dataset occurs instead between the $|200\rangle$ and $|020\rangle$ outcomes. This imbalance is now attributed to a small port-asymmetry of the effective two-port transfer matrix of the photon-number-resolving stage MZI_3 . As discussed in the Supplementary, such an asymmetry can remain compatible with standard single-port, intensity-based classical calibration [6] and is therefore not

revealed at the single-photon level, yet it directly modifies the phase-sensitive two-photon coefficient combinations governing bunching probabilities, leading to $P_{200} \neq P_{020}$ while leaving the remaining outcomes, including $|110\rangle$, in close agreement with theory.

To illustrate this effect, we direct the reader to Figs. S9 and S10 of the Supplementary Information (also reproduced in this response document (Figs. 3) and 4). These figures explicitly show how a small two-port asymmetry in MZI_3 reproduces the observed $|200\rangle$ – $|020\rangle$ imbalance for symmetric effective inputs, and why the corresponding antisymmetric NOON component remains largely unaffected, fully consistent with our experimental observations.

We have chosen to present these plots in the Supplementary Information rather than as an inset or additional panel in the main manuscript, as the corresponding main-text figure is already densely populated with experimental data and theoretical curves. Moreover, the strict limits on the number of figures in the main manuscript preclude the inclusion of a separate figure dedicated to this auxiliary analysis, which we therefore provide in full in the Supplementary for clarity and completeness.

Reviewer Comment 2.20 — Reference hygiene. Where citing software (e.g., QOptCraft and the modified interferometer package), include version tags/commit hashes.

Reply: We have revised the manuscript to explicitly specify the software versions used for all numerical simulations and interferometer decompositions. In particular, the NOON-state predictions were performed using `qoptcraft` v2.0.0, and the interferometer decomposition was implemented using a modified version of the `interferometer` Python package v1.1.1. Both packages were installed via `pip`, and the corresponding version information is now stated explicitly at the points where the software is introduced in the Methods section.

Since the software was installed from versioned `pip` releases rather than from a version-controlled repository, explicit commit hashes are not applicable. The modified version of the `interferometer` package used in this work will be submitted alongside the manuscript to ensure full reproducibility.

Reviewer 3

Reviewer Comment 3.0 — I co-reviewed this manuscript with one of the reviewers who provided the listed reports. This is part of the Nature Communications initiative to facilitate training in peer review and to provide appropriate recognition for Early Career Researchers who co-review manuscripts.

Reply: We thank the Reviewer for the careful reading of the manuscript and for the constructive comments and suggestions. We have carefully addressed the comments of all Reviewers throughout the revised manuscript and Supplementary Information, and we believe that these revisions have improved the clarity, presentation, and overall quality of the work.

Reviewer 4

Reviewer Comment 4.0 — The manuscript by Krishna et al. investigates Coherent Perfect Absorption (CPA) in the quantum regime by implementing a reconfigurable photonic network

capable of probing quantum interference effects in lossy systems. The authors design and experimentally realize an integrated beam splitter with tunable losses, transmission, and reflection coefficients on a silicon-on-insulator platform. The effective absorption is controlled via adjustable coupling to an auxiliary waveguide, enabling dynamic tuning of the system’s dissipation properties. The chip’s functionality is verified using both single- and two-photon input states. Importantly, the experiments demonstrate a clear advantage of two-photon interferometric measurements over single-photon approaches, thereby highlighting a potential metrological application of the proposed platform.

Assessment of Impact: The study addresses a timely and conceptually relevant topic at the interface of quantum optics, integrated photonics, and non-Hermitian physics. The implementation of reconfigurable CPA on a silicon photonic chip represents a significant step toward control of coherent loss mechanisms in quantum systems. The experimental validation using quantum states of light adds substantial weight to the manuscript.

Support of claims: The authors provide a thorough description of how the beam splitter parameters are implemented within the integrated circuit and convincingly demonstrate the tunability of these parameters in the experiments.

Reply: We thank the reviewer for the careful and detailed assessment of the manuscript. All points raised in the subsequent comments have been addressed through targeted revisions to the main text, figures, and Supplementary Information. In particular, we have clarified the physical interpretation of the experimental results, corrected and expanded the data analysis where required, improved figure labeling and cross-referencing, and added explicit methodological details to enhance clarity and reproducibility. Each individual comment is addressed in detail below, with corresponding changes clearly indicated in the revised manuscript.

Reviewer Comment 4.1 — The authors provide a thorough description of how the beam splitter parameters are implemented within the integrated circuit and convincingly demonstrate the tunability of these parameters in the experiments. The main claim that “all previously reported quantum CPA effects, including deterministic absorption, anti-coalescence, photon bunching and probabilistic two-photon absorption” can be observed within this architecture is supported for deterministic absorption, photon bunching, and probabilistic two-photon absorption. However, anti-coalescence of photons cannot be supported by the data shown in Fig. 5 (at $\varphi = 0.5\pi$ or 1.5π in Fig. 5b (Now 6b)). Anti-coalescence, as first reported by Vest *et al.* [10] [12 (in main manuscript)], becomes more pronounced with increasing loss. In stark contrast, the authors identify the low-absorption case as evidence for anti-coalescence.

Reply: The work of Vest *et al.* [10] addresses two-boson interference on a single lossy beam splitter and provides an important reference for understanding coalescence and anti-coalescence in the presence of loss. However, the apparent difference between their observations and ours arises from the fact that the two experiments probe different interference mechanisms and therefore cannot be directly compared at the level of the measured signatures.

In [10], the input state consists of one boson in each input port of the beam splitter (a $|1, 1\rangle$ -type input), and the interference effect is observed by scanning the relative temporal delay between the two particles. The resulting Hong–Ou–Mandel dip or peak thus reflects the degree of temporal overlap at the beam splitter, as evidenced by the millimetre-scale width of the observed features.

By contrast, our experiment is designed to emulate coherent absorption of quantum light, for which the defining signature is a phase-controlled redistribution of multiphoton probability amplitudes. We

therefore employ a path-encoded two-photon NOON-state input and vary the relative input phase, rather than the temporal delay. As discussed in Supplementary Section S7, a $|1, 1\rangle$ input does not exhibit coherent absorption in this sense, whereas the NOON-state input enables phase-dependent access to bright and dark absorption channels. This leads to interference fringes with a much finer period that are governed by the input phase.

Within this framework, anti-coalescence is clearly observed in our data. For the symmetric (Type 2) beamsplitter configuration, the probability of detecting one photon in each signal output ($|110\rangle$) exhibits pronounced maxima at relative input phases $\phi \approx \pi/2$ and $3\pi/2$ at $|A|^2 = 0$, while the same-mode two-photon probabilities in the signal outputs ($|200\rangle$ and $|020\rangle$) are simultaneously suppressed (Fig. 6(a,b) in main manuscript). These phase points correspond to clear antibunching in the signal modes and constitute the standard operational signature of two-photon anti-coalescence, analogous to the Hong–Ou–Mandel peak reported in [10], but occurring here as a function of input phase rather than temporal delay.

In addition, the fact that the loss channel is implemented as a directly measurable ancilla mode allows us to identify additional interference points at which all two-photon bunched events are suppressed. Specifically, at relative input phases $\phi = 0$ and π , and for $|A|^2 = 0.5$, in both the Type 1 (Fig. 5(a,d) in main manuscript) and Type 2 (Fig. 6(a,d) in main manuscript) beamsplitter configurations, the probabilities of finding two photons in any single output mode ($|200\rangle$, $|020\rangle$, or $|002\rangle$) approach zero. At these phases, the two photons always emerge in different output modes, distributed across the two signal ports and the ancilla port, although not necessarily in the same pair of modes.

This behaviour represents a generalized form of anti-coalescence in a three-mode setting: interference suppresses all amplitudes corresponding to both photons occupying a single mode, forcing the photons to emerge separately. The ability to directly resolve the ancilla output therefore enables a more complete identification of such anti-coalescence-like behaviour than is possible in experiments where loss channels are inaccessible.

Reviewer Comment 4.2 — Consequently, these data cannot distinguish genuine anti-coalescence from the trivial case of Hong–Ou–Mandel interference in a beam splitter with full transmission/reflection and are more indicative of the latter scenario. A suggestion for experimental evidence is provided in the Methodology section.

Reply: As clarified in our response to Comment 4.1, the experiment reported here does not probe conventional Hong–Ou–Mandel (HOM) interference of a $|1, 1\rangle$ input via temporal delay. Instead, we investigate phase-controlled interference of a two-photon NOON-state input in an effective lossy beam splitter, where the defining observable is the phase-dependent redistribution of multiphoton amplitudes rather than temporal indistinguishability.

In this context, the observed anti-coalescence cannot be attributed to a trivial case of HOM in a lossless beam splitter with full transmission or reflection. In particular, the loss channel is implemented as a directly measurable ancilla mode, allowing access to the complete three-mode output statistics. Even at finite absorption ($|A|^2 = 0.5$), we identify phase points (e.g. $\phi = 0$ and π) in both the Type 1 and Type 2 configurations (Figs. 5 and 6) where the probabilities of all two-photon bunched states, $|200\rangle$, $|020\rangle$, and $|002\rangle$, are simultaneously suppressed, while the photons always emerge in different output modes. Such behaviour cannot occur in a unitary HOM experiment, where probability is conserved within the signal modes and loss channels are inaccessible.

We therefore conclude that the data distinguish genuine anti-coalescence associated with coherent absorption from trivial HOM interference, with the observed behaviour arising from phase-controlled

interference between symmetric and antisymmetric two-photon amplitudes that selectively couple to absorbing and non-absorbing collective modes via the ancilla-assisted lossy beam-splitter transformation, rather than from temporal indistinguishability at a unitary beam splitter.

Reviewer Comment 4.3 — Another claim that remains unsupported concerns scalability. Since the title does not specify the type of quantum light, the authors should clarify why measurements with single- and two-photon states are sufficient to represent the general framework.

Reply: We thank the reviewer for this comment regarding the scope of the work and the wording of the title. In response, we have revised the title to explicitly specify the class of quantum states addressed in this study, which now reads “*Programmable coherent absorption of Fock-state quantum light in an integrated photonic circuit.*” This change is intended to clarify that the experimental demonstrations focus on discrete-variable quantum states of light.

The experiments reported with single- and two-photon input states are meant to serve as representative and physically nontrivial demonstrations of the proposed framework, rather than as an exhaustive exploration of all possible quantum states of light. The input states employed in our work—single-photon Fock states and two-photon NOON states are themselves genuine nonclassical quantum states and constitute the minimal experimentally accessible quantum resources for probing coherent absorption beyond the classical regime.

Our platform implements a programmable, ancilla-assisted realization of general passive non-unitary linear optical transformations, and its scalability follows from the structure of these transformations rather than from the specific photon number used in the experimental demonstration. Single-photon measurements establish phase-controlled coherent absorption at the level of first-order quantum interference, while two-photon experiments constitute the minimal setting in which genuinely multiphoton quantum effects can be observed.

As discussed in the main text and Supplementary section S4, higher-photon-number quantum states undergoing the same linear transformation are governed by the same scattering matrix, with output statistics determined by the corresponding many-body interference of creation operators. We therefore use single- and two-photon quantum states as experimentally accessible test cases that validate the programmable coherent-absorption framework at both the single-particle and multiparticle levels, without implying that all possible quantum states are explicitly demonstrated.

Reviewer Comment 4.4 — Additionally, it remains unclear how the resources required to implement tunable loss scale when extending the system to a 3×3 network (or larger). The authors should discuss whether the necessary number of auxiliary waveguides or the constraints of a planar geometry impose limits on scalability or reconfigurability.

Reply: We thank the reviewer for this question regarding the extension to larger lossy interferometric networks and the associated auxiliary-waveguide requirements. As also clarified in replies to the comments 2.1 and 2.7 of reviewer 2, the ancilla resources required for ancilla-assisted emulation are determined by the structure of the *net* non-unitary transformation, rather than solely by the number of individual lossy beam-splitter units.

When multiple lossy 2×2 beam splitters are combined *in series* on the same pair of signal modes, the overall transformation remains 2×2 . In this case, the ancilla requirement is fixed by the rank of the composite loss matrix $I - T^\dagger T$: one ancilla mode suffices for port-symmetric (CPA-type) loss, while two ancilla modes are required for general asymmetric loss, independent of the number of stages.

For a *multiport* network, such as a Clements-type rectangular mesh assembled from $k = N(N-1)/2$ lossy 2×2 units, the net transformation T_{net} acts on N signal modes. The minimal unitary embedding then acts on $N + m_{\text{anc}}$ modes, with

$$m_{\text{anc}} = \text{rank}\left(I - T_{\text{net}}^\dagger T_{\text{net}}\right) \leq N,$$

as discussed in Supplementary Sec. S1.2. Thus, even for general lossy multiport transformations, the number of auxiliary waveguides scales at most linearly with the number of signal modes.

We note that in a strictly planar geometry, routing independent ancilla modes to each lossy beam-splitter unit in a large mesh may pose practical layout constraints if one attempts to implement every lossy beamsplitter unit as an individually dilated block. However, this does not limit the generality of the approach: the same physical operation can be implemented by first constructing the overall non-unitary matrix T_{net} corresponding to the full network (obtained by multiplying the constituent lossy beam-splitter matrices along the appropriate modes), and then embedding this *net* transformation into a larger unitary using the same ancilla-assisted procedure demonstrated in this work. In this formulation, the required ancilla resources are set solely by $\text{rank}(I - T_{\text{net}}^\dagger T_{\text{net}})$, independent of the internal decomposition of the network.

Finally, the resulting unitary embedding on $N + m_{\text{anc}}$ modes can be realized using standard planar nearest-neighbour interferometer layouts (e.g. Clements-type meshes), so that the fundamental scaling of ancilla resources is fully captured by the rank criterion rather than by planar routing considerations.

To address these point explicitly, we have added a brief scaling remark in the last paragraph of the “Quasiunitary decomposition scheme” subsection in the Methods section of the main manuscript clarifying the dependence of ancilla resources on the net transformation, and last paragraph of Supplementary Sec. S1.2 to discuss extended lossy networks, including the constraints of planar geometry and the construction of a net non-unitary matrix T_{net} for ancilla-assisted embedding.

Reviewer Comment 4.5 — Methodology and Reproducibility: While the general experimental procedures and data presentation are clear and appear reproducible, several methodological aspects require clarification: For the calculation of the Bhattacharyya coefficients, the summation indices are not explicitly defined. Presumably, they run over the set of basis states, but this should be stated unambiguously.

Reply: We thank the reviewer for pointing this out. We have now explicitly defined the summation index in the Bhattacharyya coefficient and clarified that the sum runs over the six-dimensional two-photon Fock basis used in the analysis. This clarification has been added to the main text in the definition of Eq. 8 (of main text).

Reviewer Comment 4.6 — Typically, quantum interference effects such as photon bunching and anti-coalescence are characterized through Hong–Ou–Mandel–type experiments, contrasting the quantum interference of indistinguishable photons with a classical baseline of distinguishable ones. This distinction is particularly relevant to the manuscript’s claim of observing anti-coalescence, which currently lacks sufficient experimental evidence or discussion.

Reply: As clarified in our responses to Comments 4.1–4.3, the bunching and anti-coalescence effects reported in this work do not correspond to a Hong–Ou–Mandel (HOM) interference experiment. While both involve quantum interference of multi-photon probability amplitudes, the physical setting explored

here is fundamentally different: our experiment employs phase-controlled discrete-variable quantum input states and probes their evolution through a non-unitary, coherently absorbing linear optical transformation, rather than two photons interfering on a lossless (unitary) 50:50 beam splitter.

The effective two-port coherent absorption operation is realized by embedding a non-unitary scattering matrix into a larger unitary dilation that includes ancillary loss modes. Two-photon output probabilities therefore depend on interference between amplitudes associated with both the accessible modes and the loss channels, with the resulting redistribution between coalescence and coincidence outcomes governed by the input-state coherence, circuit parameters, and absorption strength, and not by standard HOM-type interference.

The reviewer’s question concerns the appropriate reference against which the observed anti-coalescence should be interpreted. In contrast to HOM experiments, where photon distinguishability provides a natural classical baseline, the control parameter in our experiment is the relative phase of the input quantum state, which does not modify photon distinguishability and thus cannot be used to access such a baseline experimentally.

Instead, to identify an appropriate reference for interpreting the observed anti-coalescence, we consider input states with different phase coherences within the same two-photon Hilbert space. In addition to the phase-coherent two-photon NOON state, we analyze the $|11\rangle$ Fock-state input, which consists of indistinguishable photons but does not possess NOON-type phase coherence. The $|11\rangle$ state therefore serves as a physically well-defined reference for isolating the role of phase-dependent two-photon interference in the coherently absorbing transformation.

To make this distinction explicit, we perform a theoretical analysis of mixtures of an ideal NOON state and the $|11\rangle$ Fock-state input based on the theoretical device model described in the Methods section. Figs. 8 and 9 presented in this review response document show that increasing the $|11\rangle$ contribution progressively suppresses phase-dependent modulation of the output probabilities, leaving finite, phase-independent backgrounds. The corresponding Fisher-information analysis, Fig. 10, shows a monotonic reduction of phase sensitivity with increasing $|11\rangle$ contribution, vanishing in the pure $|11\rangle$ limit. These results show that, at the phase values where anti-coalescence is observed, the output distributions corresponding to the $|11\rangle$ input provide a clear reference baseline against which the phase-dependent behaviour of the NOON-state input can be directly compared.

Reviewer Comment 4.7 — The consequences of constraining the full eight-parameter space of a general 2×2 transfer matrix to only three effective parameters $|t|$, ϕ_{rt} , and $|A|$ are not addressed. The authors should elaborate on how/if this restriction limits the accessible CPA conditions and what physical insights might be lost or preserved.

Reply: We thank the Reviewer for raising this conceptual point. A fully general complex 2×2 transfer matrix,

$$T = \begin{pmatrix} a & b \\ c & d \end{pmatrix}, \quad a, b, c, d \in \mathbb{C},$$

contains eight real degrees of freedom. The reduction to three effective parameters in this work follows from applying physically well-defined constraints appropriate for a reciprocal lossy beam splitter exhibiting coherent absorption.

First, an overall complex phase multiplying T is physically irrelevant, as it does not affect output probabilities or interference observables. Removing this global phase reduces the number of independent parameters from eight to seven.

Second, we restrict attention to *reciprocal* optical systems. Reciprocity is a fundamental property of linear, passive, time-independent, non-magnetic optical scatterers and applies to bulk beam splitters, absorbing films, and the effective two-port device emulated here. In scattering-matrix form, reciprocity enforces equality of the cross-coupling amplitudes,

$$T_{12} = T_{21}.$$

Using the conventional scattering notation, the most general reciprocal lossy beam splitter can therefore be written as

$$T = \begin{pmatrix} r_1 & t \\ t & r_2 \end{pmatrix},$$

which already reduces the parameter space by two real degrees of freedom.

Third, we impose *port symmetry*, meaning that the two external signal ports are physically indistinguishable and couple identically to the loss channel. While port symmetry is not strictly required for the mathematical existence of coherent absorption, it isolates the simplest coherent absorption regime, in which absorption can be understood in terms of collective interference of the two ports. Port symmetry enforces

$$r_1 = r_2 \equiv r,$$

leading to

$$T = \begin{pmatrix} r & t \\ t & r \end{pmatrix}.$$

The transformation is then fully characterized by three physically meaningful parameters: $|r|$, $|t|$, and their relative phase $\phi_{rt} = \arg(r) - \arg(t)$. For a passive device, these are conveniently reparameterized as $(|t|, \phi_{rt}, |A|)$, with $|A|^2 = 1 - |r|^2 - |t|^2$.

To clarify what changes without port symmetry, consider the most general *reciprocal* two-port matrix $T = \begin{pmatrix} r_1 & t \\ t & r_2 \end{pmatrix}$. Coherent absorption requires a non-trivial input vector $\mathbf{v} \neq 0$ such that $T\mathbf{v} = 0$, i.e. $\det T = r_1 r_2 - t^2 = 0$. When this condition holds, the corresponding perfectly-absorbed input generally has an *unequal* amplitude ratio $v_2/v_1 = -r_1/t$, so the CPA excitation is typically unbalanced and does not admit the simple symmetric/antisymmetric collective-mode picture. Imposing port symmetry ($r_1 = r_2$) makes the CPA eigenvectors fixed to the balanced superpositions $(1, \pm 1)/\sqrt{2}$, with eigenvalues $\lambda_{\pm} = r \pm t$, so that coherent absorption corresponds to tuning one eigenvalue to zero ($\lambda_{\pm} = 0$). Thus, restricting to the port-symmetric manifold does not remove coherent absorption; rather, it selects the canonical and most transparent regime in which CPA is realized by balanced two-port interference, enabling continuous and closed-form tunability with a minimal set of parameters.

To clarify this point for the reader, we have added a brief statement in the main manuscript (Results, 'Implementation of Non-Unitary transformation', immediately following Eq. (1)) explaining that port symmetry is not required for coherent absorption in general, but is adopted here to isolate the simplest regime, without loss of generality

Finally, in the manuscript we use an equivalent representation in which the same-port amplitude appears on the diagonal,

$$T = \begin{pmatrix} t & r \\ r & t \end{pmatrix},$$

which is related to the above form by a relabeling of ports. All reciprocity, symmetry, and CPA conditions are identical in the two conventions.

Reviewer Comment 4.8 — What is the motivation of only discussing two types of beam splitters?

Reply: The two beam-splitter classes considered in this work were chosen to exemplify two fundamentally distinct and complementary pathways for tuning coherent absorption in a port-symmetric lossy beam splitter.

Specifically, one class demonstrates absorption control via variation of the reflection-to-transmission amplitude ratio at fixed internal phase, while the other demonstrates absorption control via tuning of the internal round-trip phase at fixed amplitude symmetry. These two cases therefore highlight coherent absorption control through independent physical parameters of the beam-splitter matrix.

To make this motivation more explicit, we have added a brief clarifying sentence at the end of the paragraph in the main text describing the two beam-splitter classes, immediately preceding the “Experimental architecture” section. This paragraph already detailed the parameter constraints defining Type 1 and Type 2 beam splitters and their respective absorption-tuning mechanisms; the added sentence explicitly states that these two classes illustrate complementary amplitude- and phase-based tuning pathways.

We believe these representative cases capture the essential physics of coherent absorption in non-unitary beam splitters, and that extending the discussion to additional beam-splitter classes would not provide qualitatively new insight while significantly increasing the length of the manuscript.

Reviewer Comment 4.9 — The experimental network comprises eight waveguides, whereas the theoretical model requires only three. The manuscript should clarify the rationale for this expanded implementation — in particular, why three additional beam splitters are placed around MZIs 1–3 and how these affect the system’s overall transfer characteristics or tunability. If these additional couplers are introduced for calibration, mode matching, or symmetry reasons, this should be explicitly discussed.

Reply: We would like to clarify that the experiments were performed on a general-purpose, programmable 8-mode Clements interferometer chip (See the microscope image of the full chip presented in the Supplementary Information FigS3). This device was not fabricated specifically for the present experiment alone, but rather constitutes a universal linear-optical platform capable of implementing arbitrary interferometric transformations.

Although the full circuit comprises eight spatial modes, the coherent absorption transformation itself requires only three modes, comprising the signal modes and an ancillary loss mode used to emulate the non-unitary beam-splitter dynamics. At the output, this three-mode transformation is embedded into a six-mode subspace to enable photon-number-resolving detection, while the remaining modes of the interferometer are not required for the present implementation.

In addition, the three programmable Mach–Zehnder interferometers surrounding the central MZI 1–3 block are configured in a passive waveguide (bar) state. In this configuration, they do not perform any active interferometric operation, but simply route and connect the relevant modes between the active MZIs. This is explicitly stated in the *Experimental architecture* section of the main manuscript, where we note that these surrounding MZIs are programmed to transmit light without introducing additional transformations.

Reviewer Comment 4.10 — Aside from these points, the remaining methodological descriptions are well presented and should allow independent reproduction of the main experimental results.

Reply: We thank the reviewer for the detailed and constructive comments raised above. We have carefully addressed each of these points and have made necessary changes/additions in the revised manuscript and Supplementary Information, and we believe that the resulting clarifications and additions significantly improve the clarity, completeness, and presentation of the work.

Reviewer Comment 4.11 — The manuscript would benefit from several improvements to visual clarity and presentation consistency: Figure 1 should be expanded to illustrate the main concept of CPA—starting from the classical counterpropagating-wave picture, transitioning to propagating waves in integrated systems, and extending to the quantum regime (single- and two-photon cases). This conceptual sketch would help readers grasp the hierarchy of effects and claims.

Reply: We thank the reviewer for this helpful suggestion, which aligns closely with a similar recommendation made by Reviewer 1 (Comment 1.1). In response to these comments, we have added a new conceptual figure to the manuscript (now Figure 1) that provides a unified visual overview of coherent absorption across classical, integrated, and quantum regimes.

The revised Figure 1 starts from the classical standing-wave picture of coherent absorption with counterpropagating fields, transitions to its implementation in an integrated photonic circuit with propagating modes and an engineered loss channel, and finally extends this framework to the quantum regime, illustrating both single-photon and two-photon (NOON-state) inputs and their phase-dependent absorption behavior. This schematic is intended to clarify the hierarchy of physical concepts and experimental claims made in the manuscript, and to provide intuitive context before the detailed theoretical and experimental discussion.

For your reference, the same figure is also presented in the present review response document as Fig. 1.

Reviewer Comment 4.12 — The color scheme requires attention: the data in Figure 3 (top row) (Now figure 4) should use a consistent palette with Figure 4 (Now figure 5) to maintain visual coherence.

Reply: We thank the reviewer for pointing this out. The color palette has now been made consistent across Figures 4 (earlier Figure 3) top panel and 5 (earlier Figure 4) in the revised manuscript to improve visual coherence and readability.

Reviewer Comment 4.13 — Consider adding a conceptual comparison graphic contrasting classical and quantum CPA.

Reply: We thank the reviewer for this suggestion. A conceptual comparison between classical and quantum coherent absorption is already provided in the revised Figure 1 of the manuscript, which was introduced in response to Reviewer 1 (Comment 1.1) and further expanded following Comment 4.11.

Figure 1 explicitly contrasts the physical mechanisms governing classical and quantum CPA. Panel (a) illustrates classical coherent absorption in terms of counterpropagating waves forming a standing-wave pattern, with absorption determined by the field intensity at the absorber position. In contrast, panels (b) and (c) depict the quantum regime, where coherent absorption arises from interference of probability amplitudes associated with single- and two-photon Fock states, rather than from classical field superposition.

We note that, unlike the classical case, it is not straightforward to represent quantum coherent absorption using a direct free-space standing-wave analogue, since the underlying mechanism involves

interference between quantum probability amplitudes rather than spatial field intensities. Instead, panels (b) and (c) provide an intuitive and experimentally relevant representation of the quantum process using beam-splitter networks, illustrating how phase-controlled quantum interference governs absorption for single- and two-photon inputs.

Importantly, the integrated photonic circuit shown in panels (b) and (c) constitutes a direct analogue of a free-space lossy beam splitter, as depicted schematically in panel (a), while making the quantum interference processes explicit and experimentally accessible. Together, these panels therefore already provide a clear conceptual distinction between classical and quantum coherent absorption, as well as a unified framework linking the two regimes.

For reference, the same figure is also reproduced in the present review response document as Fig. 1.

Reviewer Comment 4.14 — In Figure 1c (Now Figure 2c), the difference between green and red shades should be more pronounced for better readability.

Reply: We thank the reviewer for this suggestion. The color contrast in Figure 2(c) (Earlier Figure 1(c)) has now been adjusted to make the distinction between the green and red shades more pronounced and improve readability.

Reviewer Comment 4.15 — The inclusion of the SPDC source in Figure 1 (Now figure 2) as part of the integrated setup contradicts Figure 2 (Now figure 3), where it is shown as an external free-space source; this inconsistency should be corrected or explained.

Reply: We thank the reviewer for pointing out this potential source of confusion. In the schematic previously shown as Fig. 1 (now Fig. 2), the SPDC source is not intended to be depicted as part of the integrated photonic circuit. In both figures, the SPDC source is an external free-space source, consistent with the experimental implementation.

In Fig. 2, we employ a simplified block-diagram representation with two open input ports as a visual aid to indicate coupling into the chip, and use arrows to explicitly show that photons generated by the external SPDC source are routed into the integrated circuit. This schematic representation was chosen for clarity and compactness, rather than to imply physical integration of the source.

To avoid any ambiguity, we have now explicitly stated in the caption of Fig. 2 that the SPDC source is external to the integrated photonic circuit. This clarification ensures consistency between the figures and accurately reflects the experimental setup.

Reviewer Comment 4.16 — The text should include explicit references to each subpanel of Figures 4 and 5 at the points where they are discussed, to enhance readability and guide the reader through the results.

Reply: We thank the reviewer for pointing this out. We have now carefully revised the manuscript to ensure that all figure panels are explicitly and consistently referenced in the main text at the appropriate locations. This has been checked throughout the manuscript to avoid any ambiguity.

Reviewer Comment 4.17 — The manuscript presents a solid experimental advance with clear relevance to integrated quantum photonics and should merit publication after major revision addressing the points raised.

Reply: We thank the reviewer for all the thoughtful and constructive feedback. We believe that the revisions made in response to all comments have significantly improved the clarity, consistency, and completeness of the manuscript, and we hope that the revised version now meets the standards for publication in *Nature Communications*.

References

- [1] A. N. Vetlugin, “Coherent perfect absorption of quantum light,” *Physical Review A*, vol. 104, p. 013716, 2021.
- [2] W. R. Clements, P. C. Humphreys, B. J. Metcalf, W. S. Kolthammer, and I. A. Walsmley, “Optimal design for universal multiport interferometers,” *Optica*, vol. 3, pp. 1460–1465, 2016.
- [3] M. Reck, A. Zeilinger, H. J. Bernstein, and P. Bertani, “Experimental realization of any discrete unitary operator,” *Physical Review Letters*, vol. 73, pp. 58–61, 1994.
- [4] N. Maring, A. Fyrrillas, M. Pont, E. Ivanov, P. Stepanov, N. Margaria, W. Hease, A. Pishchagin, A. Lemaitre, I. Sagnes, T. H. Au, S. Boissier, E. Bertasi, A. Baert, M. Valdivia, M. Billard, O. Acar, A. Brioussell, R. Mezher, S. C. Wein, A. Salavrakos, P. Sinnott, D. A. Fioretto, P. E. Emeriau, N. Belabas, S. Mansfield, P. Senellart, J. Senellart, and N. Somaschi, “A versatile single-photon-based quantum computing platform,” *Nature Photonics*, vol. 18, pp. 603–609, 2024.
- [5] J. M. Arrazola, V. Bergholm, K. Brádler, T. R. Bromley, M. J. Collins, I. Dhand, A. Fumagalli, T. Gerrits, A. Goussev, L. G. Helt, J. Hundal, T. Isacsson, R. B. Israel, J. Izaac, S. Jahangiri, R. Janik, N. Killoran, S. P. Kumar, J. Lavoie, A. E. Lita, D. H. Mahler, M. Menotti, B. Morrison, S. W. Nam, L. Neuhaus, H. Y. Qi, N. Quesada, A. Repeatingon, K. K. Sabapathy, M. Schuld, D. Su, J. Swinarton, A. Száva, K. Tan, P. Tan, V. D. Vaidya, Z. Vernon, Z. Zabaneh, and Y. Zhang, “Quantum circuits with many photons on a programmable nanophotonic chip,” *Nature*, vol. 591, pp. 54–60, 2021.
- [6] C. Alexiev, J. C. C. Mak, W. D. Sacher, and J. K. S. Poon, “Calibrating rectangular interferometer meshes with external photodetectors,” *OSA Continuum*, vol. 4, pp. 2892–2904, 2021.
- [7] S. L. Braunstein and C. M. Caves, “Statistical distance and the geometry of quantum states,” *Physical Review Letters*, vol. 72, pp. 3439–3443, 1994.
- [8] N. Tischler, C. Rockstuhl, and K. Słowik, “Quantum optical realization of arbitrary linear transformations allowing for loss and gain,” *Phys. Rev. X*, vol. 8, p. 021017, 2018.
- [9] M. Tillmann, S. H. Tan, S. E. Stoeckl, B. C. Sanders, H. D. Guise, R. Heilmann, S. Nolte, A. Szameit, and P. Walther, “Generalized multiphoton quantum interference,” *Physical Review X*, vol. 5, p. 041015, 2015.
- [10] B. Vest, M. C. Dheur, Éloïse Devaux, A. Baron, E. Rousseau, J. P. Hugonin, J. J. Greffet, G. Messin, and F. Marquier, “Anti-coalescence of bosons on a lossy beam splitter,” *Science*, vol. 356, pp. 1373–1376, 2017.

Figure 1: Classical and quantum coherent absorption in an integrated photonic analogue. (a) Classical coherent absorption at an effective lossy beam splitter. Two counter-propagating input fields interfere to form a standing wave, which can be decomposed into a superposition of two components: an absorbing mode (Mode C) that couples to the loss channel and a non-absorbing mode (Mode S) that is fully transmitted. The relative weight of these two components is controlled by the phase difference between the input fields. This standing-wave picture motivates the integrated-photonic analogue (bottom), where a balanced interferometric network maps travelling-wave inputs onto standing-wave supermodes and implements absorption by coupling only Mode C to an absorber. (b) Single-photon implementation: a single photon prepared in a coherent superposition across the two input modes undergoes single-photon quantum interference in the network, enabling phase-controlled coupling to the absorber (illustrative examples shown). (c) Two-photon implementation: two-photon quantum interference (e.g., NOON-type inputs) enables phase-dependent coherent absorption of multiphoton states (illustrative examples shown).

(a) **Verification of Eq. (2).** Analytical prediction of θ_{MZI2} versus the value extracted from quasi-unitary embedding + Clements decomposition, showing perfect correlation (100% agreement, $n = 100$).

(b) **Verification of Eq. (3).** Analytical prediction of $\phi_{MZI3} - \phi_{MZI2}$ versus the value extracted from quasi-unitary embedding + Clements decomposition, showing perfect correlation (100% agreement, $n = 100$).

(c) **Experimental validation of Eq. (2).** With the input phase set to the perfect absorption condition, the measured maximum ancilla-mode probability follows $P_{\text{ancilla}} = 2|A|^2$, yielding the same θ_{MZI2} as Eq. (2). Markers: experimental data. Line: theoretical prediction.

Figure 2: **Additional validation for Reviewer Comment 2.3.** Eqs. (2) and (3) of the main text are validated by direct agreement between the analytical formulas and the phase settings extracted from the quasi-unitary embedding + Clements decomposition used to program the chip, and by an experimental ancilla-probability validation for Eq. (2).

**Beam splitter with π -shifted reflection (Type1) - NOON state experiment -
All fock states with simulated MZI₃ asymmetry ($\lambda = 1.10$)**

Figure 3: **Type-1** ($\phi_{rt} = \pi$) NOON-state data with MZI₃-asymmetry model overlay. Measured two-photon output statistics (markers) for the six detected Fock outcomes $|200\rangle, |020\rangle, |110\rangle, |101\rangle, |011\rangle$, and $|002\rangle$ as a function of the scanned NOON phase ϕ (units of π), shown for all programmed $|A|^2$ values (color-coded). Dash-dotted curves show the corresponding theoretical prediction obtained by propagating the effective MZI₃ input state through the general two-port scattering model $S_{\text{MZI}_3} = \begin{pmatrix} t & r \\ r' & t' \end{pmatrix}$ and including the residual amplitude asymmetry parameter $\lambda \neq 1$ introduced in Eq. ???. The value $\lambda = 1.10$ (consistent with Eq. ??) is used globally for all $|A|^2$ settings. The model reproduces the observed $|200\rangle$ – $|020\rangle$ imbalance for symmetric effective inputs to MZI₃ while preserving the near-ideal antisymmetric-NOON response.

**Symmetric (Type2) beamsplitter - NOON state experiment -
All fock states with simulated MZI₃ asymmetry ($\lambda = 1.10$)**

Figure 4: **Type–2 (symmetric) NOON-state data with MZI3-asymmetry model overlay.** Same as Fig. 3, but for the Type–2 (symmetric) beamsplitter implementation. Markers show measured normalized two-photon counts versus the scanned NOON phase ϕ , for all $|A|^2$ values (color-coded). Dash-dotted curves show the theoretical prediction including the same effective MZI₃ scattering model and same residual amplitude asymmetry parameter $\lambda = 1.10$ [Eq. (S70)], which captures the bunched-output imbalance for symmetric effective inputs and leaves the antisymmetric-NOON response largely unchanged, consistent with Eqs. ??-??.

Beam splitter with π -shifted reflection (Type1) - Varied photon distinguishability analysis

Figure 5: **Theoretical analysis of two-photon output probabilities versus distinguishability for a Type 1 beamsplitter.** Normalized two-photon output probabilities as a function of the NOON phase ϕ (in units of π), shown for distinguishability values $d = 0, 0.25, 0.5, 0.75, 1.0$ and for $|A|^2 = 0.5$ and $|A|^2 = 0$. Increasing distinguishability suppresses the phase-dependent modulation across all output channels, approaching a phase-independent distribution for $d = 1$.

Symmetric beamsplitter (Type2) - Varied photon distinguishability analysis

Figure 6: **Theoretical analysis of two-photon output probabilities versus distinguishability for a Type 2 beamsplitter.** Same analysis as Fig. 5, but for the symmetric beamsplitter implementation. The interference fringes are progressively reduced with increasing distinguishability and vanish in the fully distinguishable limit.

Figure 7: **Theoretical analysis of Fisher information under varying fraction of distinguishability.** (a) Total Fisher information $F_{tot}(\phi)$ versus NOON phase ϕ for the Type 1 beamsplitter. (b) Same as (a) for the Type 2 beamsplitter. (c) Maximum Fisher information (optimized over ϕ) as a function of distinguishability for both beamsplitter types and for $|A|^2 = 0.5$ and $|A|^2 = 0$. Increasing distinguishability leads to a reduction of phase sensitivity, vanishing in the fully distinguishable limit.

Beam splitter with π -shifted reflection (Type1) - $|11\rangle$ + NOON state mixture input

Figure 8: **Theoretical analysis of the effect of $|11\rangle$ admixture on two-photon output probabilities for the π -phase-shifted (Type 1) beamsplitter.** Phase-dependent two-photon output probabilities calculated theoretically for a coherent–incoherent mixture of an ideal NOON state and a $|11\rangle$ Fock-state input, shown for several $|11\rangle$ mixture fractions and for $|A|^2 = 0.5$ (top rows) and $|A|^2 = 0$ (bottom rows). Increasing the $|11\rangle$ contribution continuously suppresses phase-dependent interference fringes, leaving a finite, phase-independent background determined by the $|11\rangle$ input statistics.

Symmetric beamsplitter (Type2) - $|11\rangle$ + NOON state mixture input

Figure 9: **Theoretical analysis of the effect of $|11\rangle$ admixture on two-photon output probabilities for the symmetric (Type 2) beamsplitter.** Same theoretical analysis as Fig. 8, but for the symmetric beamsplitter implementation. The gradual disappearance of phase-dependent modulation with increasing $|11\rangle$ admixture illustrates the transition from coherent two-photon interference to a phase-incoherent baseline set by the $|11\rangle$ input.

Figure 10: **Theoretical Fisher-information analysis under increasing $|11\rangle$ admixture.** (a,b) Total Fisher information calculated from the theoretical output probabilities as a function of the NOON phase ϕ for the Type 1 and Type 2 beamsplitters, respectively, shown for different $|11\rangle$ mixture fractions and for $|A|^2 = 0.5$ and $|A|^2 = 0$. (c) Maximum Fisher information, optimized over ϕ , as a function of the $|11\rangle$ mixture fraction. Increasing $|11\rangle$ admixture leads to a monotonic reduction of phase sensitivity, vanishing in the limit of a pure $|11\rangle$ input.